# Antiviral HIV-1 SERINC restriction factors disrupt virus membrane asymmetry

Susan A. Leonhardt[1,2,15], Michael D. Purdy[2,3,15], Jonathan R. Grover[4,15], Ziwei Yang[4,15], Sandra Poulos[2], William E. McIntire[1,2], Elizabeth A. Tatham[2], Satchal K. Erramilli[5], Kamil Nosol[5], Kin Kui Lai[6], Shilei Ding[7], Maolin Lu[4,8], Pradeep D. Uchil[4], Andrés Finzi[7,9], Alan Rein[6], Anthony A. Kossiakoff[5], Walther Mothes[4] ✉ & Mark Yeager[1,2,10,11,12,13,14] ✉

The host proteins SERINC3 and SERINC5 are HIV-1 restriction factors that reduce infectivity when incorporated into the viral envelope. The HIV-1 accessory protein Nef abrogates incorporation of SERINCs via binding to intracellular loop 4 (ICL4). Here, we determine cryoEM maps of full-length human SERINC3 and an ICL4 deletion construct, which reveal that hSERINC3 is comprised of two α-helical bundles connected by a ~40-residue, highly tilted, "crossmember" helix. The design resembles non-ATP-dependent lipid transporters. Consistently, purified hSERINCs reconstituted into proteoliposomes induce flipping of phosphatidylserine (PS), phosphatidylethanolamine and phosphatidylcholine. Furthermore, SERINC3, SERINC5 and the scramblase TMEM16F expose PS on the surface of HIV-1 and reduce infectivity, with similar results in MLV. SERINC effects in HIV-1 and MLV are counteracted by Nef and GlycoGag, respectively. Our results demonstrate that SERINCs are membrane transporters that flip lipids, resulting in a loss of membrane asymmetry that is strongly correlated with changes in Env conformation and loss of infectivity.

SERINC proteins are comprised of a family of five isoforms with 31–58% amino acid sequence identity, which are thought to incorporate serine into phospholipids[1]. The ~50 kDa integral membrane proteins have ten predicted transmembrane domains and a single N-glycosylation site (Fig. 1e and Supplementary Fig. 1a). In 2015, substantial interest in SERINCs was stimulated by the observation that the presence of host proteins SERINC3 or SERINC5 in the envelopes of HIV-1 particles reduced infectivity[2,3]. The restriction activity was highest for SERINC5, followed by SERINC3, and was not detected for SERINC2. The restriction activity of SERINC5 was counteracted by the viral protein Nef, which redirected SERINC5 to an endosomal compartment, thereby precluding incorporation into the viral envelope[4]. In addition to HIV-1,

[1]The Phillip and Patricia Frost Institute for Chemistry and Molecular Science, University of Miami, Coral Gables, FL 33146, USA. [2]Department of Molecular Physiology and Biological Physics, University of Virginia School of Medicine, Charlottesville, VA 22908, USA. [3]Molecular Electron Microscopy Core, University of Virginia School of Medicine, Charlottesville, VA 22908, USA. [4]Department of Microbial Pathogenesis, Yale University School of Medicine, New Haven, CT 06510, USA. [5]Department of Biochemistry and Molecular Biology, University of Chicago, Chicago, IL 60637, USA. [6]HIV Dynamics and Replication Program, Center for Cancer Research, National Cancer Institute, National Institutes of Health, P.O. Box B, Building 535, Frederick, MD 21702, USA. [7]Centre de Recherche du CHUM (CRCHUM), Montreal, QC, Canada. [8]Department of Cellular and Molecular Biology, University of Texas Health Science Center, Tyler, TX, USA. [9]Département de Microbiologie, Infectiologie et Immunologie, Université de Montréal, Montreal, QC, Canada. [10]Center for Membrane and Cell Physiology, University of Virginia School of Medicine, Charlottesville, VA 22908, USA. [11]Department of Chemistry, University of Miami, Coral Gables, FL 33146, USA. [12]Department of Biochemistry and Molecular Biology, University of Miami, Miami, FL 33136, USA. [13]Cardiovascular Research Center, University of Virginia School of Medicine, Charlottesville, VA 22908, USA. [14]Department of Medicine, Division of Cardiovascular Medicine, University of Virginia School of Medicine, Charlottesville, VA 22908, USA. [15]These authors contributed equally: Susan A. Leonhardt, Michael D. Purdy, Jonathan R. Grover, Ziwei Yang. ✉e-mail: walther.mothes@yale.edu; yeager@miami.edu

restriction activity of SERINC5 has been observed for other enveloped viruses including murine leukemia viruses (MLV) that express Glyco-Gag to counteract SERINCs[2,3,5,6].

SERINCs may have multiple functions that collectively interfere with virus entry. SERINC incorporation into HIV-1 particles appears to affect the conformation of the envelope glycoprotein (Env) as measured by increased exposure of sequestered epitopes, and it interferes with membrane fusion by impeding Env clustering, intermediates in the fusion pathway[7], lipid ordering[8], and expansion of the fusion pore[9–12]. Although SERINCs may impede fusion by effects on the local lipid composition in the envelope, lipid quantitation, and fractionation analysis have shown no changes in the amount and content of cell and viral membrane lipids in the presence of SERINC5[13]. Given this background of mechanistic uncertainty, we sought to determine the high-resolution structures of a human SERINC protein and ascertain the mechanism of its antiviral activity. Here, we show that cryoEM structures of hSERINC3 and AlphaFold models of hSERINC5 and hSERINC2 resemble the structure of non-ATP-dependent lipid transporters. Consistently, purified hSERINCs reconstituted into proteoliposomes induce the flipping of phosphatidylserine (PS), phosphatidylethanolamine (PE), and phosphatidylcholine (PC). In addition, our results using two lipid-flipping proteins (hSERINCs and mTMEM16F) and two retroviruses (HIV-1 and MLV) strongly suggest that lipid flipping and the associated loss of membrane asymmetry are strongly correlated with changes in Env conformation and restriction activity.

## Results and discussion
### Structure determination and molecular design
We tested a variety of recombinant systems, and the protein yield was highest for human SERINC3 expressed in Sf9 insect cells (Supplementary Fig. 2a, b). Formation of a complex with a synthetic Fab (Supplementary Fig. 2c–i) enabled the determination of a map of the 50 kDa, full-length, wild-type (WT) hSERINC3 monomer by the use of cryoEM and single-particle image analysis (Fig. 1a, b and Supplementary Figs. 3a, 4, 5). Göttlinger and colleagues presented evidence that the ability of Nef to downregulate hSERINC5 involves ICL4[4], which contains an acidic cluster motif (EDTEE) that is a proposed binding site for the clathrin adapter protein, AP2[14]. (Comparable residues in SERINC3 are SDEED, Supplementary Fig. 1b.) Hurley and colleagues demonstrated that AP2 can form a stable complex with Nef[15], and a hSERINC5-AP2-Nef complex may provide a mechanism for targeting of hSERINC5 to the endosomal/lysosomal pathway in Nef-containing HIV-1 strains[2,3,16], thereby preventing restriction by abrogating incorporation of hSERINC5 into the HIV-1 envelope. For these reasons, we also generated a cryoEM map of the ICL4 deletion mutant (ΔICL4) of hSERINC3 (Supplementary Figs. 3b, 6, 7).

The structure of WT hSERINC3 is comprised of two α-helical bundles connected by a ~40-residue, highly-tilted, "crossmember" α-helix (H4) (Fig. 1 and Supplementary Fig. 5). The average resolution of the WT hSERINC3 map was 4.2 Å (Supplementary Fig. 4d). However, the resolution of the Fab-binding bundle (Fab-proximal, H5,6,7,10) was higher (~3.6 Å), with clear definition of several tyrosine and tryptophan residues in the Fab CDR loops and the hSERINC3 epitope (Supplementary Fig. 5e, f). The map in the region of the Fab-distal α-helical bundle (H1,2,3,9) has a lower resolution (~4.4 Å) due to conformational variability between the domains (Supplementary Fig. 5a, b). The hSERINC3 maps recapitulate the high-resolution cryoEM structure of a hexameric assembly of a Drosophila SERINC ortholog (TMS1d) (Fig. 2a, b, d, e)[17]. Given that the sequence identities between hSERINC3 and hSERINC5 and hSERINC2 are 39.1 and 51.9% (Supplementary Fig. 1c), we expect that the tertiary structure is conserved in the SERINC family (Fig. 1c, d). This inference is supported by the 7.1 Å cryoEM map of hSERINC5, which shows rod-like densities consistent with α-helices in a similar disposition with the α-helical bundles present in hSERINC3[17].

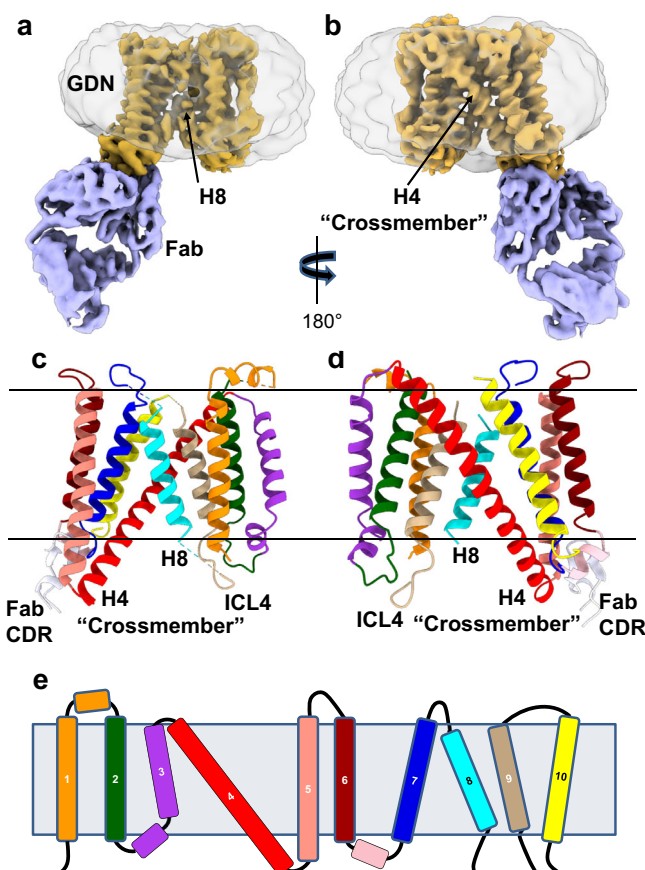

**Fig. 1 | The integral membrane protein hSERINC3 is comprised of two α-helical bundles connected by a ~40-residue, highly tilted, "crossmember" helix.** CryoEM map of full-length, WT hSERINC3 (**a**, **b**: gold) with a bound Fab (**a**, **b**: purple). The associated GDN detergent micelle is shown in transparent gray. **c**, **d** The cryoEM map shows that hSERINC3 is comprised of two α-helical bundles. The Fab-proximal bundle contains H5, 6, 7, and 10, and the distal bundle contains H1, 2, 3, and 9. The two bundles are connected by a long 40-residue, diagonal "crossmember" α-helix (H4). H4 is paired with H8, which has an ill-defined density in the full-length WT map attributed to conformational variability (**a**) and is well-ordered in the ΔICL4 deletion mutant (Supplementary Fig. 6e). **e** TM α-helices in the primary structure colored as in **c** and **d**. The horizontal lines in **c–e** demarcate the bilayer-embedded region of hSERINC3.

Furthermore, AlphaFold predicts conservation in the transmembrane architecture of SERINC3, SERINC5, and SERINC2 (Fig. 2c, g–i).

Using the Fab-proximal bundle as a reference, the distal bundle is rotated ~5° in the ΔICL4 map (Supplementary Fig. 7f) relative to the conformation in WT hSERINC3. Interestingly, the density for the H8 helix was ill-defined in the WT hSERINC3 cryoEM map (Fig. 1a), whereas H8 was well-defined in the ΔICL4 map and in TMS1d[17] (Supplementary Figs. 6e, 7c, d asterisk). We infer that the ordering of H8 and the conformational change between the helical bundles in the ΔICL4 mutant suggest that there may be allosteric communication between ICL4 and the transmembrane α-helices.

### Structural similarity of SERINC proteins with non-ATP-dependent, unregulated lipid transporters
ATP-dependent lipid transporters contain ATP-binding cassettes and are categorized as outer-to-inward flipping P4-type ATPases or "flippases" and inward-to-outward flipping ABC transporters or "floppases"[18–21]. SERINCs clearly do not have ATP-binding cassettes (Supplementary Fig. 1a). Non-ATP-dependent or unregulated lipid flipping proteins are designated as "scramblases" and exhibit both

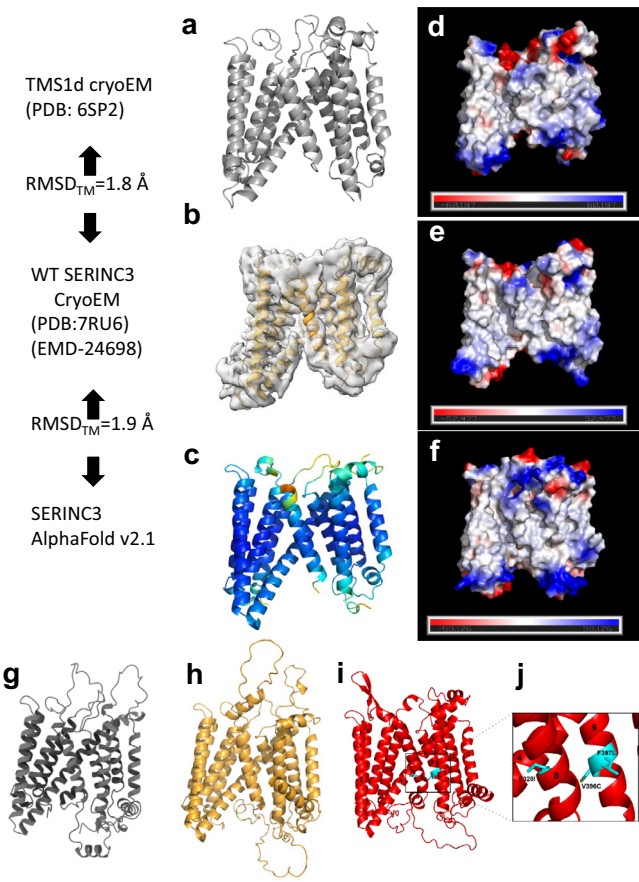

TMS1d cryoEM
(PDB: 6SP2)

RMSD$_{TM}$=1.8 Å

WT SERINC3
CryoEM
(PDB:7RU6)
(EMD-24698)

RMSD$_{TM}$=1.9 Å

SERINC3
AlphaFold v2.1

**Fig. 2 | Conservation in the molecular design of Drosophila TMS1d, hSERINC3 structures and AlphaFold SERINC models, with location of hSER-INC5 point mutants that abrogate restriction. a** The Drosophila TMS1d monomer extracted from the cryoEM structure of the hexameric protein. **b** The WT hSERINC3 cryoEM model and map. **c** An AlphaFold model of hSERINC3 colored by pLDDT confidence score (blue, very high; cyan, high; yellow, low; orange very low). Low and very low-confidence loops were the same regions missing in the cryoEM maps and were removed for comparison. Backbone RMSDs for the TM domains were calculated in PyMOL. RMSDs are lower when aligning the TM bundles separately. **d−f** Vacuum electrostatic surface potentials for each model calculated in PyMOL demonstrating fairly similar electrostatic distributions for the three models (red, anionic and blue, cationic). Ribbon representation of AlphaFold 3D models for **g** hSERINC2 (gray), **h** hSERINC3 (gold), and **i** hSERINC5 (red). Point mutations in H8 and H9 are colored in cyan and boxed. **j** Closeup of (**i**) highlighting the hSERINC5-S328I mutation in H8 and the V396C and F397L mutations in H9 that abrogate restriction. (H1 was removed for clarity.).

inward-to-outward and outward-to-inward lipid flipping. We were intrigued that hSERINC3 bears a structural resemblance with unregulated lipid transporters such as archaeal PfMATE[22] and bacterial proteins MurJ[23,24] (Supplementary Fig. 1d−f) and LtaA[25], which are comprised of two α-helical bundles connected by a long, highly-tilted crossmember α-helix. As has been proposed for the clefts between the helical bundles of PfMATE[22], MurJ[23,24], and LtaA[25], we speculate that the cleft between the helical bundles in hSERINC3 may serve as an entry portal for lipids. Consequently, we infer that all three SERINC isoforms may function as non-ATP-dependent, unregulated lipid transporters.

## Purified hSERINC3 reconstituted into proteoliposomes induces flipping of phosphatidylcholine, phosphatidylethanolamine, and phosphatidylserine

The most compelling evidence for lipid transport activity is provided by analysis of liposomes assembled from chemically defined synthetic lipids, with or without reconstitution of the putative lipid transporter.

Notably, phospholipids display minimal spontaneous flipping activity in liposomes lacking lipid transporters, water pores, detergents, or surfactants[20,26,27]. The experimental design to measure lipid flipping is depicted in Fig. 3a and is based on the work of ref. 28. A critically important control is to demonstrate that incorporation of the lipid transporter does not result in nonspecific leakiness of the liposomes (Fig. 3a, ii). This is accomplished by trapping water-soluble NBD-glucose in the liposomes containing the putative lipid transporter. In generating the liposomes, some residual NBD-glucose resides in the extravesicular space, which accounts for an initial reduction in the fluorescent signal upon the addition of dithionite, a membrane-impermeable reducing agent (Fig. 3b, 100 s time point). Importantly, the fluorescent signal reaches a stable plateau for liposomes containing a high concentration (1.5 μg/mg lipid) of hSERINC3 (Fig. 3b, 100−500 s). If the proteoliposomes had been leaky, then the signal would have continued to decline during extended incubation. Lastly, the addition of Triton X-100 solubilizes the liposomes, thereby exposing the entrapped NBD-glucose to dithionite, and the fluorescent signal fell to zero (Fig. 3b, 500 s time point). This fluorescent signal corresponds to the entrapped NBD-glucose in the nonleaky liposomes. For assessment of lipid flipping activity, proteoliposomes are generated with a fluorescent lipid (NBD-PS, NBD-PC, or NBD-PE), which is dispersed randomly in the inner and outer bilayer leaflets amongst the bulk lipids. For empty liposomes, the fluorescence signal should decrease to ~0.5 since the outer leaflet lipids will be reduced by membrane-impermeable dithionite, whereas the inner leaflet lipids will be protected from exposure to dithionite (Fig. 3c−f, blue curves). The stable fluorescence from 250−450 s demonstrates that the empty liposomes were not leaky, in which case the fluorescent signal would have been substantially <0.5. In the presence of a lipid transporter, one would expect the fluorescent signal to be reduced to near 0 since the inner leaflet lipids will be flipped to the outer leaflet where they are accessible for reduction by dithionite. In our case, we used the A$_{2A}$ adenosine receptor (A$_{2A}$AR) as a positive control, which is a known lipid transporter[29] (Fig. 3f, green curve). Our negative protein control was GltpH[30], which displayed a fluorescence decay curve (Fig. 3f, purple curve) that was close to that of empty liposomes (Fig. 3f, blue curve). In the presence of hSERINC3, the fluorescent signal of NBD-PC, NBD-PE, and NBD-PS was reduced to ~10%, which demonstrates that hSERINC3 manifests lipid flipping activity for PC, PE, and PS (Fig. 3c−e, orange curves, respectively). The reduction in fluorescence was related directly to the concentration of hSERINC3 included during liposome reconstitution (Fig. 3c). Thus, we conclude that hSERINC3 is a non-ATP-dependent, nonspecific lipid transporter for PC, PE, and PS. Nonspecific lipid flipping is a hallmark of non-ATP-dependent lipid scramblases, and we suspect that hSERINC3 flips lipids in both directions and functions as a scramblase.

## HIV-1 and MLV particles containing antiviral hSERINCs display elevated levels of phosphatidylserine, which is counteracted by Nef and GlycoGag, respectively

hSERINC5 and to a lesser extent hSERINC3 reduce the infectivity of HIV-1 particles lacking Nef (Fig. 4a). In contrast, hSERINC2 lacks antiviral activity. In addition, hSERINC5 mutations have been described that are efficiently expressed and incorporated into viral particles; however, they display varying degrees of impaired antiviral activity[17] (Fig. 4a and Supplementary Fig. 10a). PS is asymmetrically distributed across plasma membranes, with a greater fraction localized to the inner bilayer leaflet. Loss of PS asymmetry plays an important role in a number of biological processes, including apoptosis, blood coagulation, and bone mineralization[31]. For our purposes, we used the exposure of PS on viral particles as a read-out of lipid flipping, which was detected using fluorescently-labeled Annexin V. Since spontaneous flipping of PS in biological membranes is nearly zero, an increase of exposed PS would indicate the activity of lipid transporters[20,26].

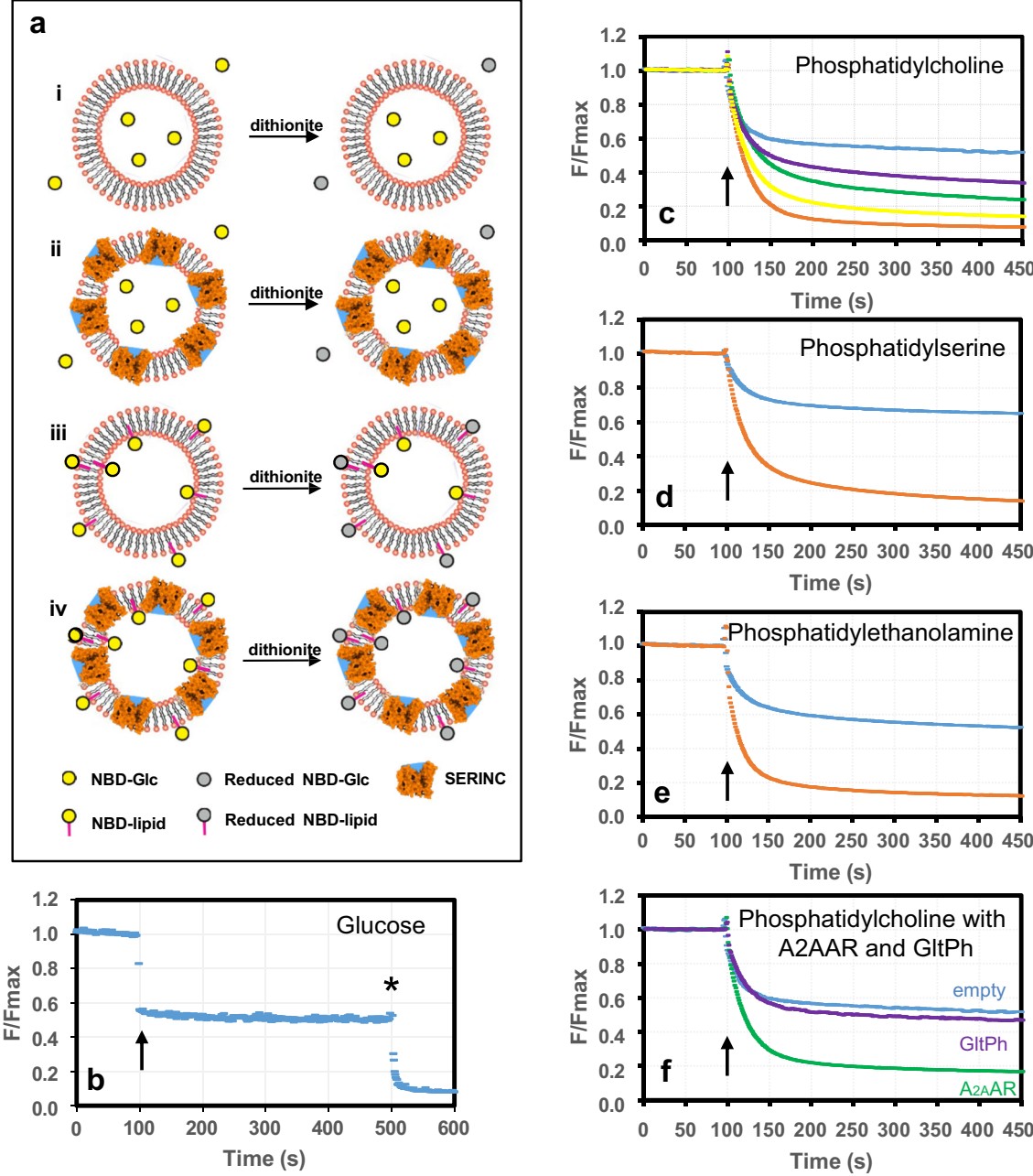

**Fig. 3 | Fluorescent proteoliposome assay demonstrates that hSERINC3 exhibits lipid flipping activity for PC, PE, and PS. a** Cartoon showing that the membrane-impermeable, reducing agent dithionite eliminates NBD fluorescence. hSERINC is displayed as being inserted randomly inside-out/outside-in, with blue showing the water cavity between the α-helical bundles. The NBD-glucose assay shows that (i) empty liposomes and (ii) hSERINC3-containing liposomes are not leaky. (iii) Exterior leaflet NBD-lipids should be accessible to dithionite resulting in ~50% reduction in fluorescence. (iv) Liposomes containing hSERINC3 should expose inner leaflet NBD-lipids for reduction by dithionite, resulting in ~100% loss of fluorescence. **b** NBD-glucose assay demonstrates that liposomes containing a high concentration of hSERINC3 (1.5 μg/mg lipid) are not leaky. The dithionite reduces the fluorescence of extravesicular NBD-glucose (60 μM). The stable fluorescent signal from 100 to 500 s indicates the protection of the encapsulated NBD-glucose from dithionite. Confirmation of entrapment was indicated by the reduction of fluorescence to near-zero upon solubilization of the liposomes at 500 s by

the addition of Triton X-100 (indicated by *). **c–f** Arrows indicate the addition of dithionite at 100 s. The blue curves correspond to the fluorescent profiles for empty liposomes, which display a 40–50% loss of fluorescence upon the addition of dithionite. **c** Representative profiles display the direct dependence of the fluorescent signal on the concentration of hSERINC3 in liposomes containing NBD-PC. Blue, purple, green, yellow, and orange curves correspond to 0.0, 0.25, 0.5, 1.0, and 1.5 μg/mg lipid of hSERINC3, respectively (assuming 100% reconstitution of protein) ($n = 2$). **d**, **e** The fluorescent signals of NBD-PS and NBD-PE, respectively, drop to near-zero upon the addition of dithionite to liposomes containing 1.5 μg/mg lipid of hSERINC3. **f** The adenosine receptor $A_{2A}AR$ (1.5 μg/mg lipid) is a known lipid transporter and serves as a positive control for the assay (green). The glutamate transfer homolog GltpH (1.5 μg/mg lipid) has previously been shown to not have lipid flipping activity and serves as a negative control for the assay (purple). **b**, **d–f** Data were representative fluorescence traces of at least three independent experiments. Source data are provided as a Source Data file.

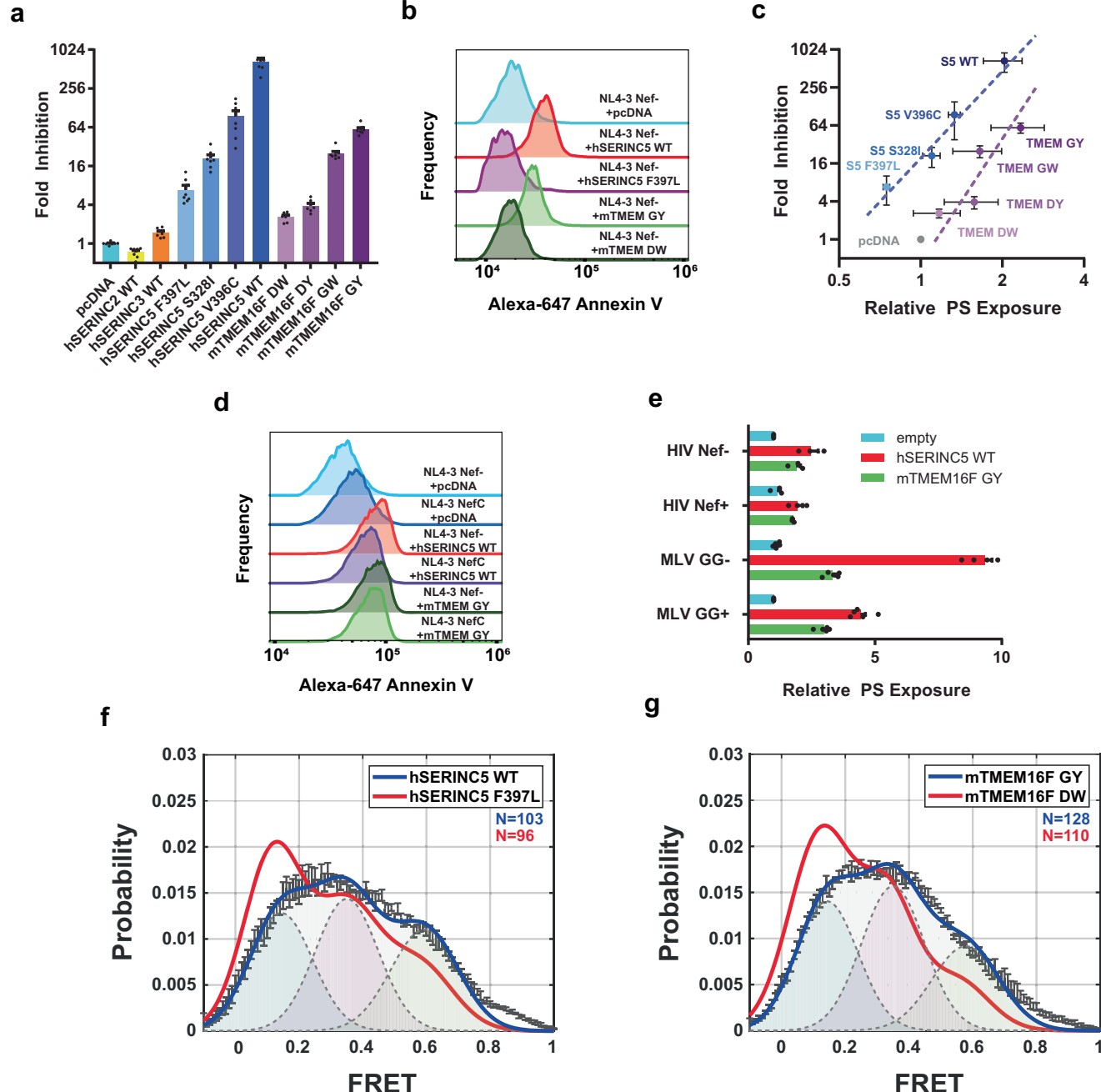

**Fig. 4 | PS exposure by hSERINC5 on virus particles correlates with inhibition of infectivity and conformational changes in Env and is antagonized by HIV-1 Nef and MLV GlycoGag. a** HIV-1$_{NL4-3}$ΔNef particles were produced in the presence of indicated plasmids and titered on TZM-Bl indicator cells by luciferase assay. Results represent the mean of $n = 7$ independent experiments ± SEM. **b** HIV-1$_{NL4-3}$ΔRTΔNef virus particles containing Gag-GFP and CD63-mRFP were produced in the presence of indicated hSERINC5 or mTMEM16F plasmid, bound to anti-CD63 magnetic beads, stained with Alexa647-annexin V, and imaged by flow cytometry. Histograms depict annexin V intensity for the GFP-positive population. Results reflect one representative experiment of $n = 3$ biological replicates. **c** PS exposure was assessed as in (**b**) and plotted in relation to virus infectivity as in (**a**) for the indicated hSERINC5 (S5) or mTMEM16F proteins. Values represent mean ± SD from $n = 3$ independent experiments. **d** HIV-1$_{NL4-3}$ΔNef or NefC-expressing virus particles were produced in the presence of indicated hSERINC5 or mTMEM16F plasmid and analyzed as in (**b**). Histograms depict annexin V intensity for the GFP-positive

population from one representative of $n = 3$ independent experiments. **e** Annexin V mean fluorescence intensity is shown for HIV-1 particles ± Nef as in (**d**). Values represent the mean ± SEM from $n = 3$ independent experiments. Percent annexin V-positive MLV particles imaged by confocal microscopy as in (Supplementary Fig. 9). Values represent the mean ± SEM from five independent experiments. **f** HIV-1$_{NL4-3}$ΔRTΔNef virus particles were produced in the presence of WT hSERINC5 or the F397L mutant and the conformational state of HIV-1 Env was analyzed by single-molecule FRET (see also Supplementary Fig. 10). N is the number of individual dynamic molecule traces complied into a population FRET histogram (gray lines) and fitted into a three-state Gaussian distribution (solid) centered at -0.15-FRET, -0.35-FRET, and -0.6-FRET. Histogram error bars represent the mean FRET probabilities ± SEM. **g** HIV-1$_{NL4-3}$ΔRTΔNef virus particles were produced in the presence of mTMEM16F GY (blue) or mTMEM16F DW (red) and the conformational state of HIV-1 Env was analyzed by single-molecule FRET as in (**f**). Source data are provided as a Source data file.

We generated HIV-1$_{NL4-3}$ΔNef virus particles in the absence or presence of hSERINC5. The incorporation of Gag-GFP was used to label virus particles. We also co-transfected the tetraspanin CD63-mRFP, which was efficiently incorporated and enabled immuno-isolation of virus particles. Two days after transfection, virus particles were harvested from the culture supernatant, bound to anti-CD63 magnetic beads, and stained with Alexa647-annexin V. Virus-bead conjugates were monitored by flow cytometry (Fig. 4b). The incorporation of hSERINC5 resulted in the exposure of PS on the surface of virus particles (Fig. 4b, c). The mutant hSERINC5-F397L (F397L is located in helix 9, Fig. 2i, j) previously identified by ref. 17 as being critical for the ability of SERINC5 to restrict HIV-1, exhibited the lowest antiviral activity (Fig. 4a) and failed to expose PS (Fig. 4b, c). Testing additional hSERINC5 mutants with varying degrees of antiviral effects[17] resulted in a direct correlation between the antiviral activity and the degree of PS exposure on the surface of virus particles (Fig. 4c).

To corroborate these findings, we used a second assay to assess PS flipping. HIV-1 particles were labeled by co-transfection of GFP-Vpr, which binds to the p6 domain of Gag and is efficiently incorporated into HIV particles. The exposure of PS was visualized microscopically by Alexa594-annexin V binding to GFP-positive viral particles immobilized on poly-lysine-coated coverslips. hSERINC3 and hSERINC5 enhanced PS exposure, while hSERINC2 did not (Supplementary Fig. 8a, d). To confirm that this result was not caused by over-expression of exogenous hSERINC proteins, we compared Gag-GFP labeled particles produced in either parental Jurkat TAg cells or hSERINC3/5 knockout Jurkat TAg cells[3,11,32]. Virus particles produced in parental Jurkat cells showed significantly higher levels of PS exposure than those produced in hSERINC3/5 knockout cells (Supplementary Fig. 8b, e).

The retroviral accessory proteins Nef and GlycoGag prevent the incorporation of SERINCs into HIV and MLV, respectively, and rescue infectivity[2,3,5,6]. Correspondingly, the exposure of PS on virus particles was abrogated when HIV-1$_{NL4-3}$ expressed the Nef gene from a clade C isolate of HIV-1 (Fig. 4d). Similarly, PS exposure was increased in xenotropic MLV, which incorporates an envelope glycoprotein (Env) that is sensitive to SERINCs (Fig. 4e). PS exposure was largely reversed when the MLV expressed GlycoGag (Fig. 4e and Supplementary Fig. 9a, c). Collectively, these data demonstrate that in two retroviruses, HIV-1 and MLV, SERINC incorporation reduces infectivity and exposes PS, which is counteracted by the retroviral accessory proteins Nef and GlycoGag, respectively.

## An active mutant of the phospholipid scramblase TMEM16F exposes PS on the surface of HIV-1 and MLV and reduces virus infectivity

To fortify the concept that lipid flipping on particles correlates with antiviral effects, we examined the effects of an unrelated PS scramblase, murine transmembrane protein 16F (TMEM16F)[33]. TMEM16F is tightly regulated and only activated during cellular processes such as apoptosis by elevated intracellular calcium levels[34]. During HIV-1 infection TMEM16F in target cells is activated through its calcium-regulated pathway to facilitate membrane fusion[35]. To investigate a possible detrimental role of PS exposure on the surface of the virus, we utilized a constitutively active murine variant (mTMEM16F)[33,36], which contains an insertion of 21 amino acids in the amino-terminal tail and a D430G point mutation (designated mTMEM16F GY). The mutant overcomes calcium regulation, displays enhanced PS scrambling activity, and is constitutively active[36]. Similar to SERINCs 3 and 5, HIV-1 and MLV containing mTMEM16F GY displayed reduced infectivity (Fig. 4a, c and Supplementary Figs. 8f, 9d) and increased PS exposure (Fig. 4b, c, and Supplementary Figs. 8c, f, 9a–c). We also assessed the functional consequences of partially active or inactive mTMEM16F mutants (see Methods for details)[37]. In contrast to the fully active mTMEM16F GY mutant, the mTMEM16F GW mutant displayed modestly reduced levels of PS exposure and infectivity inhibition (i.e., partially active), while the mTMEM16F DW and DY mutants displayed severely reduced levels of PS exposure and infectivity inhibition (i.e., inactive) (Fig. 4a–c). Taken together, these results demonstrate a correlation between PS exposure and infectivity inhibition for both hSERINC5 and mTMEM16F variants (Fig. 4a–c). As expected, the retroviral accessory proteins Nef and GlycoGag failed to efficiently counteract the effects of mTMEM16F GY (Fig. 4d, e and Supplementary Fig. 9c, d). This also concurs with the observation that wild-type TMEM16F is not active in the plasma membrane under normal physiological conditions and is not incorporated into virus particles (Supplementary Fig. 10a), and is thus not evaded by retroviruses. The finding that mTMEM16F GY remained active against ecotropic MLV, which evolved to be resistant to hSERINC5, may relate to the inability of viruses to have evaded TMEM16F (Supplementary Fig. 9c, d). Importantly, the expression of SERINCs and TMEM16F was not toxic to cells, and the absence of effects on cell viability is in agreement with previous results[36] (Supplementary Fig. 10b).

## hSERINC5 and mTMEM16F GY elicit changes in the conformation of the HIV-1 Env trimer

Incorporation of hSERINC5 into HIV-1 particles exposes CD4-induced epitopes on Env glycoprotein trimers such as the membrane-proximal external region (MPER)[32]. We used (1) single-molecule FRET (smFRET) and (2) a virus capture assay to explore this observation. smFRET has revealed that HIV-1 Env resides in a prefusion state (designated State 1), which opens in response to CD4 through a necessary intermediate (State 2), into the CD4-bound, open conformation (State 3)[38,39]. The virus capture assay gives a readout for the binding of conformation-specific antibodies[40]. The incorporation of hSERINC5 resulted in a redistribution of the conformational landscape from State 1 to the State 2 and State 3 conformations (Fig. 4f and Supplementary Fig. 10f, j). hSERINC3 exhibited only a slight change in the conformational landscape, and hSERINC2 exhibited no effect (Supplementary Fig. 10d, e, j). Correspondingly, the virus capture assay showed that hSERINC3 and, to an even greater extent, hSERINC5 increased access to CD4-induced epitope recognized by 19b (Supplementary Fig. 10k). The exposure of CD4-induced epitope recognized by 17b was observed for SERINC5, but not the F397L mutant (Supplementary Fig. 10k). The gradual increase in the exposure of antibodies 17b and 19b epitopes in response to incorporation of hSERINC5 mutants correlated with their ability to expose PS and inhibit HIV-1 (Fig. 4c and Supplementary Fig. 10k). Moreover, mTMEM16F GY, but not mTMEM16F DW similarly exhibited a profound effect on Env conformation (Fig. 4g and Supplementary Fig. 10h–j). These data illustrate that antiviral SERINCs and TMEM16F proteins also affect the conformation of the HIV-1 Env trimer, which presumably occurs as a result of a disturbance in the lipid bilayer associated with lipid transport.

## Purified WT SERINC5 and SERINC2, as well as SERINC5 mutants F397L and S328I, also flip lipids

Unlike the action of hSERINC5 on virus particles, neither hSERINC5 mutants F397L and S328I (positions in TM9 and TM8, respectively, shown in Fig. 2i, j) nor hSERINC2 enhance the exposure of PS, affect Env conformation or reduce infectivity. We also used the proteoliposome assay to assess the lipid flipping activity of wild-type hSERINC5, the mutants F397L and S328I, as well as hSERINC2. Interestingly, all proteins exhibited similar lipid flipping activities (Supplementary Fig. 11c). Although the hSERINCs exhibited similar lipid flipping, fluorescence decay curves (Fig. 4c–e and Supplementary Fig. 11c), quantitative analysis (Supplementary Fig. 11d) showed that there are differences in the flipping rates attributable to the fast component of dithionite reduction: hSERINC3 flips fastest ($5.84 \pm 0.15 \times 10^{-2}\,s^{-1}$), hSERINC5 flips the slowest ($1.92 \pm 0.06 \times 10^{-2}\,s^{-1}$), and hSERINC2 flips at an intermediate rate ($2.87 \pm 0.08 \times 10^{-2}\,s^{-1}$).

We realize that there is discordance between the preserved lipid flipping activity in proteoliposomes containing hSERINC2 or the hSERINC5 mutants (F397L and S328I) and the reduced lipid flipping and reduced restriction in HIV-1 particles containing hSERINC2 and the hSERINC5 mutants. The appeal of the proteoliposome assay is the elegant simplicity and chemical definition of the system. The liposomes are formed from synthetic lipids (POPC and POPG) doped with a fluorescent lipid, and the proteoliposome membrane contains a single, purified protein species. However, the proteoliposomes do not recapitulate several features present in cell membranes and the HIV-1 envelope. For instance, the lipids are symmetrically distributed upon generation of proteoliposomes, whereas membranes display increased concentrations of PS, PE, PIPs, and sphingomyelin in the cytoplasmic leaflet, and glycolipids are concentrated in the extracellular leaflet[41]. In addition, PIP2 and PIP3 are enriched in the HIV-1 envelope compared with the plasma membrane[42,43]. The transporters (e.g., hSERINCs and TMEM16F) are symmetrically distributed in their orientation upon reconstitution, whereas the asymmetric orientation of the proteins in the plasma membrane is retained upon particle budding. In addition, HIV-1 Env and SERINCs reside in microdomains[44], which are not present in proteoliposomes. Other proteins such as the tetraspanin CD63 reside in microdomains within the envelope[45]. These distinguishing features in the HIV-1 envelope may contribute to the restriction phenotype, not present in proteoliposomes.

## A hypothetical alternating access model for SERINC-mediated lipid flipping

For the bacterial protein MurJ, lipid II is proposed to enter the protein via a lateral gate, and the crossmember α-helix 7 serves as a lever that may facilitate the flipping of lipid II to the outer leaflet (Supplementary Fig. 1e, f)[24]. For PfMate[22] and MurJ[46], lipid flipping has been proposed to involve a mechanism similar to alternating access of membrane transporters. Such a mechanism has also been suggested for LtaA[25] but without experimental evidence. For MurJ, an alternating access mechanism is supported by multiple X-ray structures[24] and crosslinking experiments[46]. However, MD simulations of MurJ did not show the flipping of lipid II. This is not surprising given the slow time scale of lipid flipping compared with the microsecond time scales of MD simulations[19,21].

The cryoEM maps of hSERINC3 show that the helical bundles may move as rigid bodies (Fig. 5a–d) with a pivot point in the center of the crossmember helix H4. We used AlphaFold analysis to examine possible conformational changes of hSERINC5. In fact, the top two conformational states of hSERINC5 (Fig. 5e–h) displayed conformations consistent with the lever-arm-like movement of helix 7 in the MurJ lipid transporter (Supplementary Fig. 1d–f, more readily appreciated in Supplementary Movies 1 and 2)[24]. In the WT hSERINC3 cryoEM map, H8 is disordered (Fig. 1a), whereas the rod-like density for H8 is well-defined in ΔICL4-hSERINC3 (Supplementary Fig. 7c, d asterisk). Similarly, H8 has lower confidence scores in the hSERINC5 AlphaFold models, and there is a large difference in the conformation of H8 in the top two models. These observations suggest that H8 dynamics are integral to SERINC conformational changes and support a role for ICL4 in the regulation of these structural states. Our cryoEM maps of hSERINC3 (Fig. 5a–d) and AlphaFold analysis of hSERINC5 (Fig. 2i) suggest that hinge-like movements of the H4 crossmember helix enable rigid-body rotations of the helical bundles as expected for lipid flipping mediated by an alternating access model, which awaits experimental validation.

In summary, the evidence supporting the hypothesis that SERINC-mediated lipid flipping is relevant to the mechanism of restriction includes the following: (1) The architecture of hSERINC3 (Fig. 1) resembles that of other non-ATP dependent lipid transporters such as PfMATE[22], MurJ[23,24], and LtaA[25]. (2) Our comparison of the full-length and ΔICL4-constructs of hSERINC3 shows the rigid-body rotation of

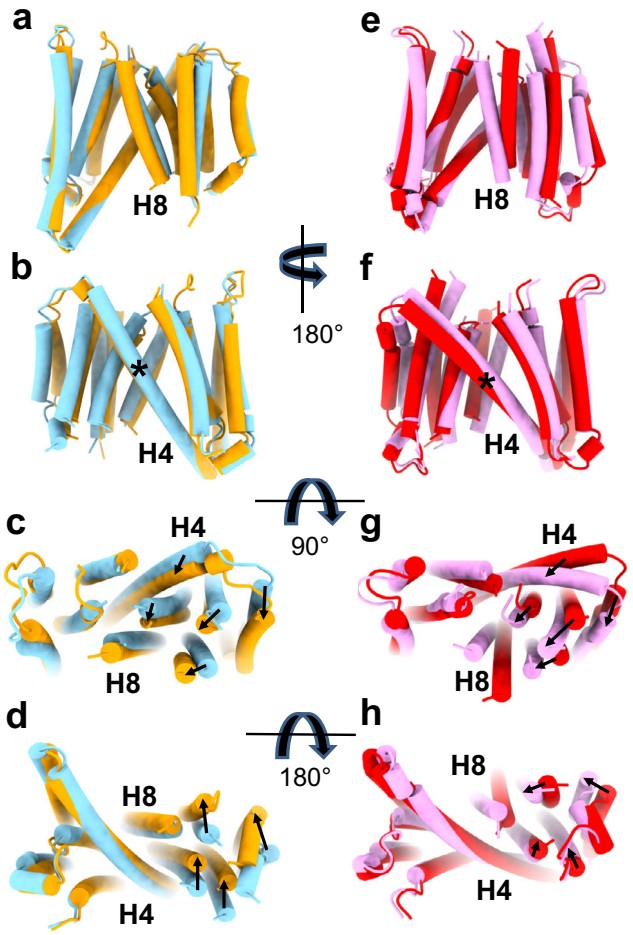

**Fig. 5 | hSERINC3 cryoEM and hSERINC5 AlphaFold analysis reveal conformational states consistent with an alternating access mechanism for lipid flipping. a–d** Superposition of full-length, WT hSERINC3 (gold) and ΔICL4-hSERINC3 cryoEM models (blue) shows a rotation of the helical bundles around H8. **e–h** The top two AlphaFold models of hSERINC5 also reveal conformational states similar to those of the hSERINC3 cryoEM structures. The domain rotations seen in the cryoEM structures and AlphaFold models involve a hinge-like flexion of the H4 crossmember helix in the center of the bilayer (*) consistent with an alternating access mechanism. In the WT hSERINC3 cryoEM map, H8 is disordered, whereas the rod-like density for H8 is well-defined in ΔICL4-hSERINC3 (Supplementary Fig. 7e). Similarly, H8 has lower confidence scores in the hSERINC5 AlphaFold models, and there is a large difference in the conformation of H8 in the top two models. These observations suggest that H8 dynamics are integral to SERINC conformational changes and support a role for ICL4 in the regulation of these structural states.

the helical domains and bending of the H4 "crossmember" helix (Fig. 5a–d). These features are similar to the conformational changes of MurJ in the flipping of Lipid II (Supplementary Fig. 1e, f)[24]. (3) Purified hSERINC3 incorporated into proteoliposomes exhibits flipping activity for PC, PE, and PS (Fig. 3c–e). Nonspecific lipid flipping is a hallmark of non-ATP-dependent lipid scramblases. (4) Annexin binding to HIV-1 particles assembled in the presence of hSERINCs3 and 5 (but not 2), indicates that PS, which is normally enriched on the inner bilayer leaflet, is exposed on the outer leaflet of the viral membrane (Fig. 4b, c and Supplementary Figs. 8a, d, 9a–c). (5) Virus particles produced in parental Jurkat cells showed significantly higher levels of PS exposure than those produced in hSERINC3/5 knockout cells (Supplementary Fig. 8b, e). (6) The accessory proteins Nef and GlycoGag, known to prevent SERINC from being incorporated into HIV-1 and MLV viral particles, respectively, reduce PS levels on the virus surface (Fig. 4d, e

and Supplementary Fig. 9a, c). (7) SERINC5 mutants with impaired antiviral activity display reduced PS exposure (Fig. 4a–c). (8) The constitutively active mTMEM16F GY mutant is incorporated into viral particles, exposes PS on virus particles and reduces virus infectivity comparable to that of SERINC5 (Fig. 4b, c) (9) SERINC and TMEM16F proteins with antiviral activity alter the conformation of Env (Fig. 4f, g and Supplementary Fig. 10f, i–k).

PS is asymmetrically distributed across plasma membranes, with increased concentration in the cytoplasmic leaflet. Therefore, exposure of PS on viral particles was a convenient read-out of lipid flipping and loss of membrane asymmetry. Our results using two lipid flipping proteins (hSERINCs and mTMEM16F) and two retroviruses (HIV-1 and MLV) suggest that lipid flipping and thereby loss of membrane asymmetry is strongly correlated with changes in Env conformation and restriction activity.

## Methods

### Constructs and expression of human SERINC proteins
**SERINC3**. The gene that encodes human SERINC3 (Genscript-OHu02717D) was inserted upstream of a thrombin protease cleavable linker (LVPRGS) and Strep II epitope (WSHPQFEK) in a modified pFastBac vector by In-Fusion cloning (Clontech). Mutagenesis using the QuikChange Site-Directed Mutagenesis kit (Agilent) was performed to change the thrombin site to a Tobacco Etch Virus (TEV) protease site (ENLYFQ\S) to facilitate expression and purification. QuikChange mutagenesis was also used to remove hSERINC3 amino acids 366-391 from this vector to generate ΔICL4-hSERINC3. For synthetic Fab production, the original hSERINC3 pFastBac plasmid had the thrombin protease linker and Strep II epitope deleted, and a Flag epitope inserted (DYKDDDDK) by site-directed mutagenesis. These constructs were expressed in Sf9 insect cells using the Bac-to-Bac Baculovirus system (Invitrogen). The cells were infected with baculovirus at 27 °C for 48 h before harvesting.

**SERINC5**. The construct design, expression, and purification of human SERINC5 recapitulated that of SERINC3 as they were done simultaneously. The gene that encodes human SERINC5 (OHu11910D) was purchased from GenScript. SERINC5 point mutations were generated by QuikChange site-directed mutagenesis.

**SERINC2**. Isoform 1 of human SERINC2 (GenScript-OHu23082D) was cloned into pFastBacI with the TEV and STREP cleavage and affinity tags upstream of the gene encoding hSERINC2. For this purpose, we used the SERINC3 construct, in which the SERINC3 open reading frame was removed by restriction enzyme digestion. The hSERINC2 gene was then inserted via ligation after PCR amplification to insert the appropriate restriction site on the 5' and 3' ends.

### Formation of hSERINC3 nanodiscs for generation of Fabs
Sf9 cell pellets infected with virus-encoding C-terminally FLAG-tagged hSERINC3 were lysed in 50 mM HEPES, pH 7.5, 50 mM NaCl, and 0.5 mM EDTA and protease inhibitors (cOmplete Ultra Tablet (Roche)). After lysis, the mixture was incubated with 2.5 mM MgCl$_2$ and benzonase (0.5 μl/ml) (EMD Millipore, Corp) for 10 min before centrifugation at 125,000×$g$ for 40 min to collect the membranes. The membranes were washed twice by homogenization in 50 mM HEPES, pH 7.5, 1 M NaCl, and 0.5 mM EDTA and were then collected by centrifugation at 125,000×$g$ for 40 min. Membranes were suspended in 40% glycerol, 10 mM HEPES, pH 7.5, 20 mM KCl, and 10 mM MgCl$_2$ before flash-freezing in liquid nitrogen and storage at −80 °C for further use.

hSERINC3 was extracted from purified membranes using 1% *n*-dodecyl-β-D-maltoside (DDM) (Anatrace), 0.2% (w/v) cholesteryl hemisuccinate (CHS) (Anatrace) in the presence of 2.0 mg/ml iodoacetamide and purified by M2 anti-Flag (Sigma) immunoaffinity chromatography. After washing with high salt buffer (1 M NaCl) and

progressively lowering the DDM/CHS concentration, hSERINC3 was eluted in a buffer consisting of 50 mM HEPES pH 7.5, 150 mM NaCl, 0.02% DDM, 0.004% CHS, and 0.2 mg/ml Flag peptide (Bio Basic, USA). β-mercaptoethanol (βME) was added to 2.0 mM before the sample was concentrated using a Vivaspin 50 kDa molecular weight cut-off filter (GE Healthcare.) The monomeric fractions were purified by size exclusion chromatography using a Superose 6 Increase column in 50 mM HEPES pH 7.5, 150 mM NaCl, 0.02% DDM, 0.004% CHS, and 2 mM βME. Before incorporation into nanodiscs, a PD-10 column (GE Healthcare) was used to remove the βME, and the hSERINC3 was concentrated to ~42 μM. Purified hSERINC3, biotinylated membrane scaffold protein MSP1D1 (provided by the Kossiakoff lab), and soybean polar lipid extract (Avanti Polar Lipids) were mixed at a molar ratio of 1.0:2.5:300. After 1 h on ice, the sample was subjected to detergent removal with Bio-Beads SM-2 (Bio-Rad). Bio-beads were removed, and the reconstitution mixture was cleared by centrifugation prior to injection onto a Superose 6 Increase column for removal of empty nanodiscs. Peak fractions were evaluated by SDS-PAGE and silver-staining (4–20% Mini-Gel; Bio-Rad) to confirm the presence of both hSERINC3 and MSP1D1. Peak fractions were pooled, adjusted to 5% with glycerol, concentrated, and flash-frozen for storage at −80 °C until use.

### Phage display
hSERINC3 was reconstituted into nanodiscs using biotinylated MSP1D1, which was chemically biotinylated and assessed for pull-down efficiency as previously described in refs. [47–49]. Five rounds of biopanning were performed using Fab Library E[50,51] in a buffer containing 25 mM HEPES, pH 7.4, and 150 mM NaCl with 1% BSA (Selection Buffer) using a method adapted from published protocols[47,48]. In the first round, biopanning was performed manually using 200 nM of hSERINC3-MSP1D1 nanodiscs immobilized onto Streptavidin paramagnetic beads (Promega). Following three washes with Selection Buffer, the beads were directly used to infect log-phase *E. coli* XL-1 Blue cells, and the phage pool was amplified overnight as described in ref. [48]. To increase selection stringency, four additional rounds of sorting were performed semi-automatically using a Kingfisher magnetic beads handler (Thermo Fisher Scientific), with the amplified phage pool from each preceding round used as input. For each of these rounds, target concentrations were decreased as follows: second round: 100 nM; third round: 50 nM; fourth round: 50 nM; and fifth round: 25 nM. The phage pools for rounds two to five were precleared with 100 μL of streptavidin particles, and for all rounds between 1.0 and 1.5 μM, empty, non-biotinylated MSP1D1 nanodiscs were used as soluble competitors. The fourth and fifth round phage pools were also tested against immobilized 25 nM empty, biotinylated MSP1D1 nanodiscs and streptavidin beads alone to evaluate hSERINC3-specific enrichment. For rounds two to five, bound phage particles were removed by elution with 1% Fos-choline-12 as described in ref. [47].

### Single-point phage ELISA
Single-point phage ELISA was used to evaluate individual clones from rounds four and five. All ELISAs were performed in 96-well plates (Nunc) coated with 2 μg/mL Neutravidin and blocked with Selection Buffer containing 1% BSA. Colonies of *E. coli* XL-1 blue harboring phagemid were used to inoculate 2xYT media supplemented with 100 μg/mL ampicillin and 10$^9$ pfu/mL M13-KO7 helper phage. The phage were amplified overnight in deep well blocks at 37 °C with shaking at 280 RPM. Amplified phage were diluted tenfold into Selection Buffer and assayed against hSERINC3-loaded 1D1 nanodiscs or empty biotinylated nanodiscs at 25 nM concentration. ELISA was performed as described in ref. [48] using an HRP-conjugated anti-M13 monoclonal antibody (GE Healthcare) and a TMB substrate kit (Thermo Fisher) to detect bound phage. Wells containing buffer alone were also used to determine the specificity of binding.

## Fab expression and purification

hSERINC3-specific binders from phage ELISAs were selected based on signal/background ratios[47] and sequenced at the University of Chicago Comprehensive Cancer Center DNA Sequencing facility. Unique clones were sub-cloned in pRH2.2 (a gift of S. Sidhu) using the In-Fusion Cloning Kit (Takara) and sequence-verified. Fab expression vectors were then transformed into *E. coli* BL21-Gold competent cells (Agilent), and Fabs were expressed as described[47,48]. Cells were harvested by centrifugation, and the cell paste was frozen until use. Fabs were purified as previously described in refs. 47,48.

## Purification of hSERINC3 and hSERINC3-Fab complex formation

Strep-tagged hSERINC3 or ΔICL4-hSERINC3 were extracted from purified membranes by incubation for 2–3 h at 4 °C in 1% DDM, 0.2% (w/v) CHS in the presence of 2.0 mg/ml iodoacetamide in 50 mM HEPES, pH 7.5, 300 mM NaCl, and 2.5% glycerol. The unextracted material was removed by centrifugation at 125,000×g for 40 min. The supernatant was incubated overnight at 4 °C with Strep-Tactin Beads (Qiagen), typically using 0.75 ml packed beads per two liters of original culture volume. After binding, the beads were washed with high salt buffer (1.0 M NaCl) containing a progressively lower DDM/CHS concentration (0.05% DDM/0.001% CHS-final). The beads were then exchanged into a high salt buffer (500 mM NaCl) containing 0.5% glyco-diosgenin (GDN, (Anatrace)) and incubated with PNGaseF (New England Biolabs) for 1 h at room temperature. The GDN concentration was lowered by one wash in high salt buffer (500 mM NaCl) containing 0.1% GDN, and hSERINC3 was eluted from the beads with desthiobiotin (Sigma) in low salt buffer (150 mM NaCl) containing 0.0045% GDN. The desthiobiotin was removed using a G-25 buffer exchange column (GE Lifescience). The hSERINC3 concentration was determined and mixed with a 1.25 molar excess of the Fab on ice for 1 h before the addition of the same Strep-Tactin beads used previously, but which had subsequently undergone regeneration. The bead mixture was incubated overnight at 4 °C under gentle agitation. The beads were subsequently washed with high salt buffer (1.0 M NaCl), followed by low salt buffer (150 mM NaCl), with both buffers containing 0.0045% GDN. hSERINC3 was eluted in low salt buffer containing 2.5 mM desthiobiotin and then concentrated using a centrifugal filter (Vivaspin, 50 kDa MWCO (GE Healthcare)). The solution was subjected to a 5 min spin at 17,000×g in a refrigerated microcentrifuge before separation on a Superdex 200 10/300 column (GE Healthcare) equilibrated with SEC buffer (50 mM HEPES, pH 7.5, 150 mM NaCl, and 0.0045% GDN). Peak fractions were subjected to silver stain analysis to confirm the presence of both hSERINC3 and the Fab. Selected monomer peak fractions with the Fab were pooled and concentrated.

## CryoEM grid preparation and data collection

All grids for cryoEM were prepared in the University of Virginia Molecular Electron Microscopy Core (MEMC). C-Flat 1.2/1.3 400C grids (Protochips) were glow discharged in a Pelco EasiGlow for 45 s at 20 mA. A 2.1 µL aliquot of WT hSERINC3-Fab at ~4 mg/mL was applied to the carbon side of each grid in a Vitrobot Mark IV held at 4 °C and 95% relative humidity. Grids were blotted in the Vitrobot with Whatman #1 filter paper, using a blot force of 2 and blot time of 7 s. The WT SERINC3 sample was then vitrified in liquid ethane cooled by liquid nitrogen. ΔICL4-SERINC3-Fab was crosslinked with 10 mM BS3 for 45 min at room temperature, concentrated to ~4 mg/mL, and cryoEM grids were prepared in the same way, with a blot force of 5 and blot time of 7 s. Suitable grids and grid regions of hSERINC3-Fab samples were identified by screening atlas images on the UVa MEMC Titan Krios (Thermo Fisher Scientific) using the Falcon 3EC detector. CryoEM data collection of the best grids was performed at the New York Structural Biology Center (NYSBC) Titan Krios III equipped with a Gatan K2 detector and a GIF energy filter. A priority for cryoEM of hSERINC3-Fab was to record images of the thinnest ice that still contained a suitable number of particles. The optimal ice thickness was 30–40 nm. In ice 25–30 nm thick, particle exclusion was significant, and in ice thinner than 25 nm, all particles were excluded. In ice thicker than 50 nm, the Leginon maximum resolution estimates degraded substantially. Leginon data collection criteria were set to exclude imaging positions with ice thickness less than 25 nm and greater than 60–70 nm using the Leginon Holefinder ice thickness determination method.

All other hSERINC3-Fab and ΔICL4-hSERINC3-Fab cryoEM datasets were collected on the UVa MEMC Titan Krios equipped with a Falcon 3EC detector using EPU automated acquisition software (Thermo Fisher Scientific). Data collection details are summarized in Supplementary Table 1.

## CryoEM image processing and reconstruction

All cryoEM image processing and reconstructions were performed using RELION 3.0 or RELION 3.1. An overview of the processing is presented here for the wild-type hSERINC3-Fab data collected at NCCAT. Processing was similar for the other datasets, and RELION flowcharts and results for the WT SERINC3-Fab and ΔICL4-hSERINC3-Fab are presented in Supplementary Figs. 3–7. Movie frame alignment and dose weighting were performed with MotionCor2 v1.1[52] using 5 × 5 patches. In the case of the NCCAT dataset, images were 2x binned for a final pixel size of 1.298 Å/pixel. Contrast transfer function (CTF) estimation was performed with CTFFIND 4.1.13 on the aligned and dose-weighted micrographs. Images that met the following CTFFIND criteria were selected for particle picking: maximum resolution estimates better than 3.8 Å, underfocus range −0.8 to −2.2 µm, and astigmatism <200 Å. Particles from a subset of the data were picked automatically using a Laplacian of Gaussian (LoG) model with minimum and maximum diameters of 120 and 180 Å, respectively. Following extraction, these particles were "cleaned" using 2D classification, the resulting particle set was used to generate an ab initio initial model, and then subjected to 3D classification. The best 3D model was used for particle auto-picking from the full dataset. Picked particles (515,230) were extracted with a box size of 200 pixels (259.6 Å) (Supplementary Fig. 4a). The particle set was "cleaned" again using 2D classification with 24 classes. Some 2D classes clearly showed α-helices for hSERINC3, while particles were poorly aligned in other classes. All classes that appeared to include a substantial number of "real" particles were selected for further processing, resulting in 495,552 particles. The same model used for particle picking was low-pass filtered to 15 Å and used for 3D classification with six classes and a spherical mask of 180 Å. 3D classification resulted in one good class with 164,700 particles. Particles from this class were subjected to 100 iterations of 3D classification (one class) with the same spherical mask and a regularization parameter of T = 20. The resulting map was low-pass filtered to 15 Å and used to generate a mask with a soft edge of 11 pixels. The mask and the map from 3D classification (low-pass filtered to 4 Å) were used for half-map 3D auto-refinement with an initial angular sampling of 0.9° and an initial offset range of 1 pixel. Aligned particles were subjected to CTF refinement and Bayesian polishing and refined as before with no noticeable improvement in the GSFSC resolution or map appearance. Resolution values of the final maps were determined by the gsFSC (gold-standard FSC) method[53,54] where the FSC curve crossed a correlation value of 0.143.

A significant obstacle to a high-resolution refinement of WT hSERINC3-Fab and ΔICL4-hSERINC3-Fab maps was apparent conformational heterogeneity between the two bundles of transmembrane α-helices. We tried a variety of masking, particle subtraction, and local refinement strategies and programs (RELION 3.0/3.1[55], cryoSPARC v2.1[56], and cisTEM 1.0[57]) in attempts to improve the maps. Alignments were dominated by the Fab-proximal helical bundle and the Fab variable domains (designated Fab_v), which contributed to the reduced resolution of the Fab-distal helical bundle. However, attempts to refine the hSERINC3 helical bundles individually failed due to the

small size of each fragment (e.g., Fab-distal helical bundle <20 kDa). In order to generate the best map for each hSERINC3 helical bundle and measure the approximate resolution of each, we generated separate maps for the proximal bundle-Fab$_v$ domain and the distal bundle. The separate maps were used to create a mask for each domain, which were then used for post-processing in RELION. Optimal B-factor sharpening values were determined empirically. B-factors of −75 Å$^2$ for the Fab-proximal domain and −50 Å$^2$ for the Fab-distal resulted in the best combination of map detail and continuity of density. The resulting maps were combined in Chimera and used for structure analysis and model building.

### Model building and refinement

An initial model was built in Coot 0.8.9.2[58] using the hSERINC3-Fab$_v$ domain map. Ideal α-helices were placed on the map using the topology of the *Drosophila melanogaster* hSERINC homolog TMS1d[17]. The model of the hSERINC3-Fab-proximal bundle (H4-5-6-7-10) was built using bulky side chains and proline densities, combined with MEMSAT-SVN and Protter[59] hydropathy and PSIPRED 4.0[60] secondary structure predictions to determine the sequence register in the map. Putative models of the Fab-distal bundle (H1-2-3-8-9) were built using a similar strategy combined with hydrophobicity and electrostatic analysis of each model in ChimeraX[61] and analysis of membrane insertion energies with the PPM server[62].

An initial model of the Fab$_v$ domain was generated using the Robetta server[63] and the synthetic antibody structure PDB: 5bk1[64]. The Fab$_v$ model was manually rebuilt in Coot (version 0.9.7). The hSERINC3-Fab$_v$ model including all side chains was refined against the map in real space using ISOLDE 1.0b3[65] in ChimeraX (version 1.3). During ISOLDE refinement helical restraints were used on the full length of all transmembrane helices with the exception of H3, which had two discontinuous rods of density and was therefore divided into two helical segments. We then truncated the side chains of the Fab-distal bundle to Cα in the hSERINC3 model. which we determined was most likely to be correct based on agreement with the cryoEM map and energetic and hydrophobicity analysis.

A model of ΔICL4-hSERINC3-Fab$_v$ was generated from the hSERINC3-Fab$_v$ model by adding an ideal helix to the ΔICL4-hSERINC3 model in the H8 helical density between the two bundles. The H8 helical density had insufficient side chain features to unambiguously position the sequence, so we relied on the MEMSAT-SVM and Protter predictions and energetic and hydrophobic analyses to build and position a probable model for H8. The resulting ΔICL4-hSERINC3-Fab$_v$ model was refined in ISOLDE with helical restraints and a low weight on the map contribution to the refinement simulation ($10–20 × 1000$ kJ mol$^{-1}$ (map units)$^{-1}$ Å$^3$).

Due to the absence of some side chains in the WT hSERINC3-Fab and most of the side chains in the ΔICL4-hSERINC3-Fab cryoEM maps, we used AlphaFold version 2 (10.1038/s41586-021-03819-2) to generate complete models of the hSERINC3 transmembrane domains. We believe these hybrid cryoEM/AlphaFold models are the best estimations of the hSERINC3 structures. We ran AlphaFold v2.1 on a local workstation with the full database, except for the exclusion of the TMS1d structure (PDB:6SP2). We removed low-confidence (low pLDDT scores) loops from the top-scoring AlphaFold model, which corresponded to the density that was not visualized in the cryoEM maps. All the transmembrane α-helices of the AlphaFold model had very-high confidence scores except H8, which was also weaker in the WT cryoEM density map. We docked the top AlphaFold hSERINC3 model into each cryoEM map (WT and ΔICL4) in ChimeraX, then used ISOLDE v1.2 to refine the models. To preserve the AlphaFold side chain information, we used a very low map weight ($5 × 1000$ kJ mol$^{-1}$ (map units)$^{-1}$ Å$^3$), and we used secondary structure restraints for all helical segments. Following ISOLDE refinement, we added the CDR of the bound Fab and performed further refinement of the models using Phenix Refine

(version 1.19.2-4158) and real space refinement in COOT (version 0.9.7). The deposited hybrid models (PDB:7RU6 and PDB:7RUG) include all side chains, and details are summarized in Supplementary Table 1.

### Purification of hSERINC's for incorporation into liposomes

Strep-tagged hSERINC3, hSERINC2, hSERINC5, hSERINC5-F397L, and hSERINC5-S328I were purified as described above for the hSERINC3-Fab complexes, except that the proteins were not exchanged into GDN. After elution from the Strep-Tactin Beads (Qiagen), the proteins were further purified on a Superdex 200 10/300 column (GE Healthcare) equilibrated with SEC buffer (50 mM HEPES, pH 7.5, 150 mM NaCl, and 0.05% DDM/0.001% CHS). Peak fractions were pooled and concentrated to ~1.0 mg/ml.

### Preparation of liposomes

The preparation of liposomes and proteoliposomes and the performance of the lipid flipping assay were adapted from protocols published by the Menon lab[28]. Liposomes and proteoliposomes were prepared in 50 mM HEPES, pH 7.5, and 150 mM NaCl (i.e., liposome buffer). A glass syringe (Hamilton, 50 and 500 μl), was used to add 215.5 μl POPC (25 mg/ml, in chloroform, Avanti) and 24.6 μl POPG (25 mg/ml, in chloroform, Avanti) to a 25 ml spherical flask, yielding a molar ratio of POPC:POPG = 9:1. The spherical flask was attached to a rotary evaporator (Buchi Rotavapor, speed 8), and the lipids were dried under argon gas at room temperature for ~30 min. The flask was then transferred to a vacuum desiccator overnight. The dried lipid film was hydrated with ~10 ml liposome buffer using gentle swirling, and the suspension was then sonicated at a frequency of 40 Hz for 20–30 min. To generate 400 nm unilamellar liposomes, the solution was extruded 11x using a LiposoFast Basic extruder (Avestin Inc.). The extruder membrane was then changed from a pore size of 400 to 200 nm, and the solution was extruded 5x to generate 200 nm liposomes. The phospholipid concentration of the lipid suspension was quantified before reconstitution as described below. Due to losses during extrusion, the concentration was typically ~3.6 mM. The liposomes were stored at 4 °C and could be used for ~1 week after preparation.

### Quantitation of phospholipids

Phospholipid quantitation was performed by subjecting the extruded liposomes and also proteoliposomes to oxidation by perchloric acid[28]. After cooling to room temperature, an aliquot (1 ml) of water was added followed by 400 μl of freshly prepared 12 g/L ammonium molybdate and 50 g/L sodium ascorbate. After vortex mixing the sample was heated at 100 °C for 10 min. The sample was then cooled to room temperature, and the absorbance at 797 nm was measured. Absorption standards of Na$_2$HPO$_4$ were run in parallel to generate a calibration curve.

### Reconstitution of hSERINC proteins into liposomes

Multiple reactions of 1 ml volume were typically performed simultaneously with volumes scaled accordingly. For each 1 ml reaction, reconstitution of hSERINC's, and control proteins (A$_{2A}$AR and Glt$_{Ph}$) into liposomes was performed using 2 ml plastic Eppendorf tubes. An aliquot of the extruded lipid solution (typically 800 μl) was added to 34.4 μl of 10% (w/v) DDM in liposome buffer, yielding a total volume of 840 μl. The solution was incubated for 3 h at room temperature using an end-over-end rotator. During the last hour, 9.4 μl of the NBD-lipid label (PS, PC, or PE, Avanti, Inc.) in chloroform was dried under argon and resuspended in 45 μl of liposome buffer containing 0.1% DDM. The final DDM concentration in the 1 ml reaction was ~7 mM. For the preparation of 1 ml of empty liposomes, 45 μl of the NBD-lipid label in 0.1% DDM liposome buffer was added to the swelled lipid solution (840 μl), to which 60 μl of 0.1% DDM liposome buffer was added, followed by

55 µl of detergent-free liposome buffer. For the preparation of 1 ml of *proteoliposomes*, hSERINC3, 2 and 5 (wt, F397L, S328I) in DDM/CHS was concentrated to 1.0–1.2 mg/ml using a GE Healthcare Vivaspin (100 kDa cut-off). The protein was first diluted in 0.1% DDM liposome buffer to 150 ng/µl. The reconstitution mixture was prepared by the addition of the following solutions to 840 µl of the swelled lipid mixture in the order indicated: (1) An aliquot (x µl) of the protein, (2) 45 µl of the dissolved NBD-lipid, (3) 0.1% DDM liposome buffer (x µl) and (4) 55 µl of detergent-free liposome buffer. Liposomes and proteoliposomes were incubated for an additional hour with end-over-end rotation.

For the preparation of $A_{2A}AR$ proteoliposomes, the protein was purified in a buffer containing DDM/CHS and 100 µM of the agonist adenosine (see below for details). Extruded lipids were swelled using 7.98 mM DDM for 3 h with end-over-end rotation. DDM/CHS solubilized $A_{2A}AR$, NBD-PC and adenosine were added to the final 1.0 ml reaction containing 7.0 mM DDM and 100 µM adenosine. Empty liposomes with adenosine were reconstituted in parallel.

For the preparation of $Glt_{Ph}$ liposomes, the purified protein (a kind gift from Olga Boudker (Weill Cornell)) was diluted in 0.1% DDM liposome buffer containing L-asp (1 mM) to 150 ng/µl, and the final 1.0 ml reaction included 1 mM L-asp.

Detergent removal and generation of proteoliposomes was achieved using polystyrene beads (Bio-Beads in SM-2 Adsorbent Media, Bio-Rad, Inc.). The protocol was adapted and refined from ref. 66, so that the NBD-glucose assay (see below) demonstrated minimal leakage of the liposome-incorporated fluorescent label. To prepare ~12 proteoliposome samples, 6–7 g of Bio-Beads were weighed into a conical tube and washed as follows: (1) Methanol (30 ml) was added to the conical tube, and the beads were incubated for 10 min with end-over-end rotation for 10 min, (2) After settling of the beads, the methanol was decanted, (3) Steps (1) and (2) were repeated two additional times, (4) 50 ml water was added to the beads, which were incubated as before for 10 min; (5) The water was decanted, and the beads were washed for a final 10 min in 50 ml of liposome buffer, (6) The buffer was decanted, and a glass Pasteur pipette was used to remove any residual liposome buffer, and (7) The beads were stored at 4 °C.

The 1 ml proteoliposome solution was transferred to a 2.0 ml Eppendorf tube containing 100 mg of Bio-Beads, which was incubated with end-over-end rotation for 1 h at room temperature. An additional 160 mg of Bio-Beads were then added, and the mixture was again incubated with end-over-end rotation for an additional 2 h. The proteoliposome solution was then transferred to a third Eppendorf tube containing 160 mg of Bio-Beads, which was incubated overnight at 4 °C. The following morning the solution was transferred to a fourth Eppendorf tube containing 160 mg of Bio-Beads and incubated for a final 2 h at 4 °C. Any residual BioBeads were removed by passage of the solution through a Micro Bio-Spin chromatography column (0.8 ml; Bio-Rad). The proteoliposome sample in the 2 ml Eppendorf tube was placed on ice in preparation for the lipid flipping assay.

### Quantitation of protein incorporated into liposomes

The efficiency of incorporation of protein into liposomes (at the 1.5 µg/mg of lipid concentration) was determined by fluorescent imaging of Simply Blue stained proteins separated by SDS-PAGE, using purified proteins (i.e., SERINCs, $A_{2a}AR$, and $Glt_{Ph}$) to generate a standard curve. Proteoliposomes were collected by high-speed centrifugation, solubilized with SDS sample buffer, and separated by SDS-PAGE using 4–20% gradient Mini-PROTEAN TGX gels (Bio-Rad). In order to control for liposome-related artifacts in protein quantitation, samples for standard curves were prepared by centrifugation of equivalent volumes of empty liposomes, followed by resuspension of the pellet with the addition of increasing concentrations of purified protein. After a 15-min incubation at room temperature, SDS sample buffer was added to the liposome-standard mixture prior to electrophoresis; gels

were then stained with Simply Blue and destained with water. Proteins were imaged at 700 nm using an Odyssey imager (Li-Cor), and fluorescence intensity was measured. The integrated intensity of the standard curve was analyzed using GraphPad Prism (version 9.4.1), and signals from proteoliposome samples were interpolated by linear regression analysis. Typical protein recovery for hSERINC3, hSERINC5, and hSERINC2 was $43 \pm 10\%$, $41 \pm 4\%$, and $58 \pm 11\%$, respectively, for proteoliposomes set up at the 1.5 µg/mg of lipid density ($n = 4$). Phospholipid recovery for the same proteoliposomes used for protein quantitation was $81 \pm 3\%$, $82 \pm 2\%$, and $81 \pm 1\%$, respectively. Errors were calculated using the standard error of the mean.

### Fluorescent lipid flipping assay

Measurements were performed at 23 °C using a HORIBA Jobin Yvon Fluoromax-3 Spectrofluorometer with excitation and emission wavelengths set at 470 and 530 nm, respectively, and a slit width of 5. A 2 ml sample was prepared using 20 µl of liposomes (or proteoliposomes) and 1.98 ml of liposome buffer. The solution was pipetted into a 10 × 10 mm pathlength cuvette (Hellma High Precision Cell), inserted into the sample holder of the spectrofluorometer, and stirred with a magnetic "flea" bar at speed 5. The fluorescent signal was recorded for 100 s to ensure a stable signal prior to the addition of dithionite. Dithionite is especially prone to oxidation, and aliquots (10 mg) were weighed into 1.5 ml microfuge tubes, which were kept on ice until use. During the 100 s equilibration scan, the dithionite solution was prepared by the addition of 60 µl of ice-cold 1 M Tris (pH 10) to the microfuge tube. The dithionite was easily dissolved by vortex mixing for a few sec. At 100 s, 40 µl of the dithionite solution was added to the 2 ml sample in the cuvette, yielding a final concentration of 20 mM dithionite. FluoEssence v3.9 software was used to record the fluorescence vs time measurements. The plots of fluorescence intensity displayed an artifactual vertical spike at 100 s due to light leakage while opening the lid of the sample chamber during pipetting of the dithionite solution into the cuvette. The fluorescence data were plotted on a 0 to 1 ordinate scale as normalized values, F/Fmax, where Fmax was the average fluorescence between 80–85 s.

### NBD-glucose assay

We used an assay with water-soluble NBD-glucose[67] to rule out the possibility that the reconstitution of hSERINC3 into liposomes resulted in permeability of the bilayer to dithionite, and thereby a greater reduction in fluorescence compared with empty liposomes (Fig. 3b). After the swelling step described above, an aliquot of NBD-glucose (Sigma) was added to the 1 ml reaction (instead of NBD-lipids) such that the final concentration was 60 µM NBD-glucose. In addition, to amplify any effects due to hSERINC3, the protein concentration was increased from 1.5 to 2.0 µg/mg of lipid. Although the majority of the extra-vesicular NBD-glucose was removed during the incubations with Bio-Beads, some residual external NBD-glucose accounted for the initial drop in fluorescence upon the addition of dithionite at 100 s. The fluorescent signal from 100–500 s was then due to the entrapped NBD-glucose, and the nearly constant signal indicated that the addition of hSERINC3 did not elicit permeability of the dithionite through the bilayer. Lastly, the elimination of the fluorescent signal upon addition of 1% Triton X-100 at 500 s was due to solubilization of the proteoliposomes and dithionite reduction of the entrapped NBD-glucose.

### Expression and purification of $A_{2A}AR$

An amino-terminally fused T4 lysozyme–$A_{2A}AR$ construct with $A_{2A}AR$ truncated at position 316 was used as a positive control for the flippase experiments[68]. The construct contained a FLAG epitope immediately after a hemagglutinin signal sequence at the amino terminus and a TEV cleavage site immediately following the FLAG epitope. The construct lacks the first 4 residues of the $A_{2A}AR$ amino terminus. $A_{2A}AR$ was expressed in Sf9 (*S. frugiperda*) insect cells using the Bac-to-Bac

Baculovirus Expression System (Invitrogen). Sf9 cells were grown in ESF921 media (Expression Systems) at 27 °C, diluted to a density of $2.0 \times 10^6$ cells/ml, and infected with a high-titer baculovirus stock at an MOI of 3 for 48 h, harvested by centrifugation and stored at −80 °C. Viral titers were performed by the flow cytometric method[69] on a Guava easycyte 8HT in which cells were stained with Anti-gp64-PE antibody (Expression Systems).

Membranes were prepared from receptor-infected insect cells as described in ref. 70. All steps were performed at 4 °C unless otherwise noted. Cells were resuspended in hypotonic Buffer A (10 mM HEPES, pH 7.5, 10 mM $MgCl_2$, 20 mM KCl, and EDTA-free protease inhibitor cocktail) and lysed by Dounce homogenization. Membranes were recovered by centrifugation at 125,000×$g$ and washed again with Buffer A. Two additional washes were performed with buffer A containing 1 M NaCl (Buffer B). After the final wash, membranes were weighed, resuspended in 2.5 volumes of buffer A containing 40% (v/v) glycerol (Buffer C), and flash-frozen in liquid nitrogen.

For purification of $A_{2A}AR$, frozen membranes were thawed in a tenfold excess of 20 mM HEPES, pH 7.5, 10% glycerol, 4 mM $CaCl_2$, 100 μM adenosine, and EDTA-free protease inhibitor cocktail (Buffer D) containing 300 mM NaCl. The suspended membranes were solubilized with 0.5% DDM/CHS for 4 h with gentle rotation. $A_{2A}AR$ extracts were clarified by high-speed centrifugation at 125,000×$g$, and incubated with anti-FLAG-M1 agarose affinity gel (Sigma-Aldrich, catalog number A4596) for overnight binding. Chromatography was then performed using a Bio-Rad Econo-column. The bound $A_{2A}AR$ was washed with 30 column volumes of Buffer D containing 300 mM NaCl and 0.1% DDM/CHS, followed by a wash with 30 column volumes of Buffer D containing 500 mM NaCl and 0.05% DDM/CHS. The final wash was with 20 column volumes of Buffer D containing 150 mM NaCl and 0.025% DDM/CHS. The bound $A_{2A}AR$ was eluted from the FLAG-M1 affinity resin with 10 column volumes of an elution buffer containing 20 mM HEPES, pH 7.5, 10% glycerol, 300 mM NaCl, 100 μM adenosine, 10 mM EDTA, 0.025% DDM/CHS, and EDTA-free protease inhibitor cocktail. Eluted $A_{2A}AR$ was concentrated to 500 ml in a Vivaspin 6 mL concentrator with a 100,000 molecular weight cutoff and further purified by size exclusion chromatography using a Superdex 10 ×300 column with buffer containing 25 mM HEPES, pH 7.5, 150 NaCl, 0.025% DDM/CHS, and 100 μM adenosine. Fractions containing the $A_{2A}AR$ were pooled and purified as described above, and concentrated to ~1 mg/mL for fluorescent lipid flipping experiments.

## Cells, plasmids, and reagents

HEK293 cells were a kind gift from John A.T. Young. TZM-bl cells were obtained from the NIH AIDS Reagent Program. HEK-293T cells were obtained from ATCC. Jurkat TAg and hSERINC3/5−/− Jurkat TAg cells were a kind gift from ref. 3. 293T/17 (293T) was a kind gift from Jens H. Kuhn (NIH/NIAID). HT1080-mCAT1 cells stably expressing the mCAT1, the ecotropic MLV receptor from mouse cells, have been described previously in ref. 6. HEK293 and Jurkat cells were maintained in RPMI (Gibco) media containing 10% (v/v) fetal bovine serum, 1% Penicillin/streptomycin, and 1% Glutamine. HEK-293T cells were maintained in DMEM (Gibco) media containing 10% (v/v) fetal bovine serum, 1% Penicillin/streptomycin, and 1% Glutamine. pNL4-3 Nef- was generated by replacing amino acids 31-33 of the Nef open reading frame with stop codons. pNL4-3 NefC contains the Nef gene from a clade C HIV-1 isolate in place of the NL4-3 Nef gene and was a kind gift from Heinrich Göttlinger. pNL4-3 ΔRT with V1Q3 and V4A1 tags was described previously[38]. The HIV-1 GagPol plasmid, pCMV ΔR8.2, was obtained from Addgene. The HIV-1 Env plasmid, pE7 NL4-3 Env, was a kind gift from Joseph Sodroski. The plasmids pCD-Env, expressing Moloney MLV Env; pBabe-Luc, an MLV vector expressing firefly luciferase; pRR1485, encoding MLV "Gag-Pol" with inactivated *env*; pRR1322, encoding MLV "Gag-Pol" with inactivated *glycogag* and *env*; pRR1321, expressing xenotropic MLV Env were previously described in ref. 6. A

pRR1842 plasmid expressing a GFP-tagged codon-optimized MLV *gag* was constructed as follows. The codon-optimized *gag* fragment (a kind gift of Wei-Shau Hu) was cloned into pcDNA3.1(+) (Thermo Fisher) between BamHI and NotI sites. The EGFP gene was amplified from pEGFP (Addgene) and then inserted into the 3′ end of *gag* with a linker sequence "GGAGGTGGAGCATCA" in between. The CD63-mRFP plasmid was a kind gift from Gillian Griffiths[71]. hSERINC3 and hSERINC5 open reading frames were amplified from Jurkat and TZM-Bl cDNA respectively, and inserted into pcDNA3.1(+) by standard PCR techniques. Both constructs bear C-terminal FLAG tags. SERINC5 point mutants were generated by quick-change site-directed mutagenesis. The pcDNA3.1(+) hSERINC2-FLAG plasmid was a kind gift from Felipe Diaz-Griffero. GFP-Vpr was a kind gift from Tom Hope. pNL4-3 Gag-EGFP Nef- was constructed by inserting the EGFP coding sequence and a flexible poly-linker at the C-terminus of Gag.

Transmembrane protein 16F (TEM16F) is a calcium-dependent phospholipid scramblase that consists of eight transmembrane α-helices. It is localized to the cell surface and regulates the lipid distribution across the inner and outer leaflets of the plasma membrane. It remains in an inactive state under normal physiological conditions and is only activated during cell apoptosis by the associated elevation of intracellular calcium. In its active state, TMEM16F equilibrates the phospholipid content across both leaflets of the plasma membrane by serving as an ATP-independent bidirectional lipid transporter[34]. This effectively causes elevation of the phosphatidylserine (PS) level on the outer leaflet, which is asymmetrically distributed to the cytoplasmic side in non-apoptotic cells. Here, we utilized an alternatively spliced variant form of murine TMEM16F (a kind gift from Shigekazu Nagata), which harbors an Asp-to-Gly mutation at amino acid position 409, which sensitizes TMEM16F to respond to the normal intracellular calcium concentration[33]; and a 21 amino-acid-insertion at codon 24, which increases the PS scrambling activity of the D409G (now D430G) mutant. As a result, this long-form variant murine TMEM16F exhibits a constitutively high level of PS scrambling activity in an ATP- and calcium-independent manner[36]. We also observed that the WT form of mTMEM16F is not efficiently incorporated into HIV-1 particles (Supplementary Fig. 10a). However, mTMEM16F containing the 21 amino acid-insertion at the N-terminus facilitates the incorporation into virus particles (Supplementary Fig. 10a). Given this observation, we chose to characterize 4 mutants, which exhibit varying levels of PS exposure and inhibition of infectivity, in the context of mTMEM16F containing the N-terminal insertion. The mutants are designated as mTMEM16F DW (inactive), mTMEM16F DY (inactive), mTMEM16F GW (partially active), and mTMEM16F GY (fully active). mTMEM16F DY contains the N-terminal insertion in the context of the WT protein sequence. mTMEM16F DW contains the N-terminal insertion and the 563W mutation, the combination of which severely impairs its PS scrambling activity. mTMEM16F GW contains the 430G mutation described above, as well as the 563W mutation, which partially impairs its PS scrambling activity. mTMEM16F GY contains the D430G mutation described above and the wild-type 563Y residue, which confer constitutive PS scrambling activity[37].

## Annexin V staining of retroviral particles

For HIV-1 particle staining experiments, HEK293 cells were transfected with 0.2 μg pNL4-3 Gag-EGFP ΔRT, 0.1 μg CD63-mRFP, 0.1 μg hSERINC/TMEM16F, and 0.05 μg GFP-Vpr plasmids in quadruplicate per 24-well culture plates using PEI. At 48 h post-transfection, virus particles were harvested from the supernatant, filtered through 0.45 μm filters, and incubated with anti-CD63 magnetic beads (Invitrogen) for two hrs at 4 °C. Virus-bead conjugates were washed with PBS + 0.1% BSA, and incubated with Alexa647-conjugated annexin V diluted 1:50 in annexin V binding buffer (Invitrogen) for 45 min at ambient temperature. Samples were washed with annexin V binding buffer and resuspended in annexin V binding buffer prior to analysis on a BD Accuri FACS

instrument using Accuri software (BD, version 1.0.264.21). Data were analyzed using FlowJo (BD, version 10.8.1). For conditions comparing HIV-1 ±Nef, virus particles were prepared by transfecting cells with 0.8 µg pNL4-3 Nef- or pNL4-3 NefC, 0.4 µg CD63-mRFP, 0.4 µg hSERINC/mTMEM16F, and 0.2 µg GFP-Vpr. Virus-bead conjugates were stained as described above and fixed with 4% PFA for 10 min at ambient temperature prior to analysis. For confocal-based experiments, HEK293 cells were plated in 6-well culture plates and transfected with 2 µg pNL4-3 ΔRT Nef-, 100 ng GFP-Vpr, and 100 ng hSERINC or mTMEM16F plasmids or empty vector using Fugene 6, according to the manufacturer's instructions. After 48 h, virus-containing supernatants were filtered and placed in poly-lysine-coated glass-bottom dishes (MatTek). Particles were then washed and stained with Alexa594-annexin V (Thermo Fisher Scientific) in an annexin binding buffer (10 mM HEPES, pH 7.4, 140 mM NaCl, and 2.5 mM CaCl₂). Confocal imaging was performed using a Nikon Eclipse TE2000-E microscope equipped with 444-, 488-, and 561-nm lasers, a Yokogawa CSU 10 spinning disc confocal laser scanning unit, and an Andor Zyla sCMOS camera. Images were acquired using iQ3 software (Andor, version 3.4.1). Images were quantified manually using ImageJ (NIH, version 1.52a) by drawing 20-pixel circular regions of interest around each virus particle, subtracting background fluorescence, and calculating the ratio of annexin fluorescence to GFP-Vpr fluorescence. For Jurkat-derived virus particle analysis, HEK293 cells were transfected with pNL4-3 Nef- and pNL4-3 Gag-EGFP Nef- at a ratio of 1:1. After 48 h, supernatants were filtered and used to spinoculate Jurkat cells at 1200× *g* for 2 h at room temperature in the presence of polybrene. Jurkat cells were incubated for 24 h, washed, and cultured for 4 days. Jurkat supernatants were then collected, filtered, and subjected to microvesicle depletion using anti-human CD45 magnetic beads (BioLegend) and a separation magnet (Miltenyi Biotec). Virus particles were immobilized, stained, and imaged as described above. MLV-Xeno and -Eco pseudovirions with or without GlycoGag protein were produced by transient transfection of 293T cells using Mirus TransIT-293 transfection reagent. Cells (6 × 10⁵ cells per well) in a six-well plate were co-transfected with a mixture of Env-defective MLV "Gag-Pol" with or without GlycoGag protein (3 µg of pRR1485 or pRR1322), pBabe-Luc (0.5 µg), an Env expression plasmid (0.5 µg of pCD-Env or pRR1321), pBJ5-Ser5 or pRR1839 (0.1 µg), and pUC-CMV as a filler plasmid. The GFP-tagged MLV particles were prepared in a similar way but with the addition of pRR1842 (0.15 µg) to the transfection. All supernatants were collected at 48 and 72 h post-transfection, pooled, and filtered through 0.22-µm filters. Filtered supernatant containing GFP-tagged MLV particles produced in the presence or absence of hSERINC5 or constitutively active mTMEM16F was placed in poly-lysine-coated glass-bottom dishes (MatTek). Particles were then washed and stained with Alexa594-annexin V (Thermo Fisher Scientific) in an annexin binding buffer (10 mM HEPES, pH 7.4, 140 mM NaCl, and 2.5 mM CaCl2). Confocal imaging was performed using a Leica TCS SP8 confocal laser scanning microscope (Leica, Germany). Images (*n* = 5) of each treatment were acquired by using the same laser power and digital gain parameters with the same microscope. 12-bit images with a 1024 × 1024 field size were acquired with 60x objective lenses. After acquiring, the images were processed with ImageJ and further analyzed with CellProfiler[72] to quantify the number of GFP-positive MLV particles and the number of those particles with which the annexin V signal was colocalized. Finally, the percentage of phosphatidylserine-positive MLV particles was determined. GraphPad Prism 9 (version 7.01) was used to perform a two-way or one-way analysis of variance (ANOVA) to assess statistically significant differences.

## Single-molecule Förster resonance energy transfer microscopy (smFRET)

Labeled virus particles were prepared, imaged, and analyzed as described previously in ref. 38. Briefly, HEK-293T cells were tranfected with 5.85 µg pNL4-3 ΔRT Nef- and 0.15 µg pNL4-3 ΔRT V1Q3/V4A1 Nef-, and 100 ng of the indicated hSERINC/mTMEM16F plasmid, using PEI (Polysciences). Viruses were collected at 48 h post-transfection, filtered using 0.45 µm filters, and sedimented through 15% sucrose cushions at 25,000×*g* for 2 h at 4 °C in an SW-28 swinging-bucket rotor. Virus particles were then resuspended in SFP labeling buffer, containing calcium and magnesium, and incubated overnight at ambient temperature with LD550-cadaverine and LD650-CoA (Lumidyne), Transglutaminase (Sigma), and Acyl carrier protein synthetase, AcpS (homemade). Viruses were then biotinylated with DSPE-PEG2000-Biotin lipid (Avanti Polar Lipids), and purified by sedimentation through a 6–18% continuous OptiPrep gradient at 40,000×*g* for 1 h at 4 °C in an SW41 swinging-bucket rotor. Virus particles were placed on a streptavidin-coated fused silica slide and incubated in a smFRET imaging buffer containing an oxygen-scavenging system composed of protocatechuic acid (PCA) and protocatechuate dioxygenase (PCD). Data were acquired on a home-built prism-TIRF microscope. Data were analyzed using Matlab (MathWorks, version R2017a) and a custom based analysis software package, SPARTAN (version 3.8.1)[73], courtesy of Scott Blanchard, and custom Matlab scripts, developed in the Mothes lab. Dynamic molecule traces were combined into population FRET histograms and fitted to a three-state Gaussian distribution centered at -0.15, -0.35, and -0.6 FRET.

## Virus infectivity and Western immunoblot analysis

For HIV-1 infectivity measurements, HEK293 cells were transfected with 0.2 µg pNL4-3 Nef- plasmid, 0.1 µg pHIV-In-GLuc plasmid, and 0.1 µg of the indicated hSERINC/mTMEM16F plasmid in quadruplicate per 24-well culture plates using PEI. Virus-containing supernatants were harvested at 48 h post-transfection, pooled, filtered, and used to infect TZM-Bl indicator cells. Secreted Gaussia Luciferase activity was measured 48 h post-infection. To measure the infectivity of the MLV pseudovirions, we first seeded the HT1080-mCAT1 cells in 12-well plates at 1 × 10⁵ cells per well. The following day, cells were pre-treated with 20 µg/ml DEAE-dextran (Sigma) at 37 °C for 30 min and then infected with MLV-Xeno or Eco pseudovirions that had been produced in the presence or absence of hSERINC5 or constitutively active mTMEM16F. At 48 h post-infection, firefly luciferase activity was measured in cell lysates, using the luciferase assay system (Promega) following the manufacturer's protocol. Meanwhile, the amount of input virus in the filtered virus supernatant was determined by quantitative anti-p30 immunoblotting. The specific infectivity was finally determined by normalizing the luciferase signal with the amount of virus input. For Western immunoblot analysis of HIV-1 particles, cells were transfected with 2 µg pNL4-3 Nef- plasmid and 200 ng hSERINC/TMEM16F plasmids. Virus-containing supernatants were collected at 48 h post-transfection, filtered, and pelleted by centrifugation at 20,000×*g* for 90 min at 4 °C. Cells and virus pellets were lysing for 20 min on ice in hSERINC lysis buffer[5] (10 mM HEPES, pH 7.5, 100 mM NaCl, 1 mM TCEP [Tris(2-carboxyethyl)phosphine], 1% DDM [*n*-Dodecyl-β-ᴅ-maltoside]) containing cOmplete mini protease inhibitor cocktail (Sigma-Aldrich, St. Louis, MO, USA). Lysates were mixed 1:1 with 2X LDS sample buffer containing 50 mM TCEP (Invitrogen), incubated for 5 min at ambient temperature, and subjected to SDS-PAGE. Proteins were transferred to PVDF membranes and subjected to immunoblotting with anti-HIV Ig polyclonal serum (NIH ARRRP) and anti-FLAG monoclonal antibody (Sigma-Aldrich, Clone M2). For immunoblotting of MLV particles, virions were concentrated by ultracentrifugation (25,000×*g*, at 4 °C for 1.5 h) through a cushion of 20% sucrose prepared in PBS (Invitrogen). The pellets were resuspended in PBS, and the virus samples for Western immunoblot analysis were prepared in 1 x NuPAGE LDS sample buffer containing 50 mM TCEP-HCl. To prevent hSERINC5 precipitation, samples were not heated above 37 °C. Filtered supernatant and concentrated virions underwent NuPAGE electrophoresis using 4 to 12% Bis-Tris

polyacrylamide gels (Invitrogen), followed by transfer to Immobilon-FL polyvinylidene difluoride membrane (Millipore). Membranes were blocked in Odyssey blocking buffer (Li-Cor) and probed with the following primary antibodies: rabbit anti-p30CA (NIH AIDS Reagent Program), rabbit anti-hSERINC5 (Abcam), rabbit anti-MLV gp70 (NIH AIDS Reagent Program), and rabbit anti-FLAG (Invitrogen). Secondary antibodies conjugated to DyLight 800 or 680 (Li-Cor) and the Li-Cor Odyssey imaging system were applied to specifically detect the corresponding protein. Images were analyzed using ImageStudioLite (Li-Cor).

### Cell viability measurements

To determine the viability of the virus-producing cells, 30,000 HEK293E cells were co-transfected on a 96-well plate with 14 ng of pcDNA3.1-based DNA constructs expressing either wild-type or mutant hSERINC5 or mTMEM16F, along with 28 ng of pNL4-3 Gag-EGFP Nef- and 14 ng of CD63-mRFP. Forty-eight hrs post-transfection, 100 μL of freshly thawed CellTiter-Glo One Solution Assay (Promega) reagents were added to each transfected 96 well and mixed well to induce cell lysis. The mixed content was then incubated at room temperature for 10 min, during which the released ATP from lysed cells activate the Luciferin to generate luminescent signals that reflect the total ATP level within and hence the viability of the transfected cells. Luminescence was then collected for each well on a TECAN SPARK® Multimode Microplate Reader.

### Virus capture assay (VCA)

HEK-293T cells were transfected with 3.5 μg pNL4-3 ΔVpr ΔEnv F-Luc, 1 μg VSV-G, 3.5 μg pNL4-3 Nef-, 1 μg GFP, and 200 ng of the indicated hSERINC plasmid in 10 cm tissue culture dishes. Viruses were then subjected to the VCA with the indicated antibody at a concentration of 5 μg/ml, as described previously in ref. 40.

### Statistics and reproducibility

The exponential "curve stripping" method was used to fit multiple exponential curves to the fluorescence decay data in Supplementary Fig. 11 and is described as follows. The first few seconds of the protein-free liposome data (blue) representing the quenching of fluorescence in the outer bilayer leaflet by dithionite can be fitted using a single exponential decay curve (black dashed). Subtracting the black dashed curve from the blue curve leaves a double exponential curve (black solid). This double exponential is subtracted from the observed proteoliposome decay curve, leaving the decay resulting from dithionite quenching in combination with SERINC lipid flipping (black dotted). Fluorescence decay of empty liposomes (blue) results from the sum of quenching due to dithionite reduction of lipids in the outer membrane leaflet of the liposomes (black dashed) and a slower reaction of unknown origin (black solid). Decay of proteoliposomes containing hSERINC5 (I, red), hSERINC3 (ii, gold), and hSERINC2 (iii, gray) results from the sum of the unknown slow reaction (black solid) and quenching due to dithionite reduction of fluorescence in the outer leaflet as each hSERINC flips the NBD-PC lipids (black dotted). The forward (α) and backward (β) flipping rate within each isoform is similar, supporting the assumption that there is no preferential direction for the incorporation of SERINCs into liposomes. However, the flipping rates between isoforms are different. hSERINC3 flips fastest $(\alpha = (5.84 \pm 0.15) \times 10^{-2}\,\text{s}^{-1}, \beta = (5.5 \pm 0.3) \times 10^{-2}\,\text{s}^{-1})$, hSERINC5 the slowest $(\alpha = (1.92 \pm 0.06) \times 10^{-2}\,\text{s}^{-1}, \beta = (1.38 \pm 0.08) \times 10^{-2};\,\text{s}^{-1})$, and hSERINC2 at an intermediate rate $(\alpha = (2.87 \pm 0.08) \times 10^{-2}\,\text{s}^{-1}, \beta = (2.45 \pm 0.13) \times 10^{-2}\,\text{s}^{-1})$. On average for all 3 SERINC proteins, ~77% of the fluorescence decay in proteoliposomes can be attributed to dithionite quenching of outer leaflet lipids in combination with SERINC lipid flipping, while 23% is due to other phenomena, such as leaky liposomes and/or a fraction of multilamellar liposomes.

For cryoEM, ice thickness was the critical variable for achieving the highest resolution. Even with the same experimental conditions for plunge freezing into ethane (type of grid, use of glow discharge sample volume, blot time, blot force, temperature, humidity, single-sided blotting, and Whatman filter paper without heating), we encountered variability in ice thickness. Particles were excluded from regions of thin ice and were superimposed in regions of thick ice. Consequently, the number of grids examined for all of the test constructs (~40) was the best indicator of the number of experimental replicates.

### Reporting summary

Further information on research design is available in the Nature Portfolio Reporting Summary linked to this article.

## Data availability

The atomic coordinates and EM maps for WT hSERINC3-Fab and ΔICL4-hSERINC3-Fab have been deposited in the Protein Data Bank (www.rcsb.org) and EMDB (www.ebi.ac.uk/pdbe/emdb/) with accession codes 7RU6 and EMD-24698 and 7RUG and EMD-24705, respectively. Source data are provided with this paper.

## Code availability

Scripts related to the exponential curve stripping analysis (Supplementary Fig. 11d) are available at https://github.com//eatatham/lipid_flipping. The Single-molecule Platform for Automated Real-Time Analysis (SPARTAN) software package can be obtained from Scott Blanchard at https://www.scottcblanchardlab.com/software. Custom Matlab scripts developed in the Mothes lab for smFRET data analysis are available upon request.

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

## Acknowledgements

We thank Shigekazu Nagata for the mTMEM16F plasmids, Felipe Diaz-Griffero for hSERINC2 plasmid, Heinrich Göttlinger for Jurkat TAg and SERINC3/5 knockout cells, and Tom Hope for the GFP-Vpr plasmid. We thank Olga Boudker (Weill Cornell) for providing purified GltPH. CryoEM data were primarily collected at the University of Virginia Molecular Electron Microscopy Core facility (RRID:SCR_019031), which was established via recruitments funds to M.Y., NIH grant G20 RR31199, NIH grant S10-RR025067, and NIH grant S10-OD018149. CryoEM data were also collected with the assistance of William Rice at the National Center for CryoEM Access and Training (NCCAT) and the Simons Electron Microscopy Center located at the New York Structural Biology Center, which is supported by the NIH Common Fund Transformative High Resolution Cryo-Electron Microscopy program (U24 GM129539), and by grants from the Simons Foundation (SF349247) and NY State. This work was supported by the National Institutes of Health (NIH) grants P50 AI15046 and U54 AI170856-01 (M.Y., W.M., and A.A.K.), R01 AI154092 (M.Y.), R01 GM117372 (A.A.K.), and P01 AI150471 (W.M.), by the Intramural Research Program of the NIH, National Cancer Institute, Center for Cancer Research, and in part by the NIH Intramural AIDS Targeted Antiviral Program. S.D. and A.F. were supported by the CIHR grant 352417 and a Canada Research Chair. Some molecular graphics and analyses were performed with the University of California, San Francisco Chimera package. Chimera is developed by the Resource for Biocomputing, Visualization, and Informatics at the University of California, San Francisco (supported by the National Institute of General Medical Sciences Grant P41 GM103311).

## Author contributions

S.A.L., S.P., and W.E.M. performed protein expression and purification. S.A.L., S.E., K.N., and A.A.K. generated synthetic Fab proteins. M.D.P. and S.A.L. performed cryoEM, image analysis, and model building. M.D.P. performed MD simulations and AlphaFold analysis. S.A.L. performed the proteoliposome fluorescent lipid flipping experiments with input from M.Y. E.A.T. performed "curve stripping" analysis of the fluorescence decay curves, with input from M.Y. and S.A.L. J.R.G., Z.Y., S.D., P.D.U., A.F., and W.M. performed virus imaging experiments, flow cytometry, HIV-1 capture, and infectivity experiments. M.L., J.R.G., and W.M. performed smFRET studies. K.K.L. and A.R. performed PS exposure experiments in MLV particles. The manuscript was written primarily by S.A.L., M.D.P., J.R.G., W.M., and M.Y., with input from all authors throughout experimentation and manuscript preparation.

## Competing interests

The authors declare no competing interests.
