## [Peer Review File · Nature Communications]

Antiviral HIV-1 SERINC Restriction Factors Disrupt Virus Membrane AsymmetryEditorial Note: This manuscript has been previously reviewed at another journal that is not operating a transparent peer review scheme. This document only contains reviewer comments and rebuttal letters for versions considered at *Nature Communications*. Mentions of the other journal have been redacted.

Reviewer #3 (Remarks to the Author):

The authors have improved their manuscript further by providing additional data, discussing the limit of the current studies, and revising the manuscript. I agree with the authors that there could be a number of reasons for the discrepancy between in vitro and in vitro flipping activities and authors provides a discussion for potential reasons for the discrepancy. I recommend publication of this manuscript to Nature communications.

Reviewer #5 (Remarks to the Author):
See Attached

Reviewer #5 Attachment on the following page.

I have previously reviewed this manuscript, when it was under consideration by [REDACTED]. I thank the authors for responding to my criticisms of the manuscript at this point. However, in terms of revisions to the manuscript, most notably the figures, very little has been done to address my concerns. I detail these in full below:

1. Figure 1 remains very raw in its presentation at present, I recommend:
 - a. removing the black backgrounds from the molecular graphics images.
 - b. Improving the resolution of the topology plot (Fig 1e).
 - c. Aiming towards a presentation like shown in the figure below from the same journal: <https://www.ncbi.nlm.nih.gov/pmc/articles/PMC8654953/figure/Fig1/> i.e. (A) introduce to the reader what the proposed function of SERINC3 is, (B) provide a high quality illustration of the EM density. (C) Show the cartoon representation of SERINC3 beside the EM and detail key features. (D) Map out the topology. (E) Compare structural states (or save this for a later figure).
2. Figure 2 should be SI material. The best iterations of this type of figure detail each TM helix or feature one-by-one and also provide full details on the EM workflow, e.g.: SI Fig 2 from <https://www.ncbi.nlm.nih.gov/pmc/articles/PMC8654953/> This is equivalent to Extended Figure 3 in this manuscript.

3. Figure 3c-e require legends for each trace. There are resolution issues here, with the DPI for 3e being lower than the other traces.

4. Figure 4 then jumps to three SERINC5 mutants – S328I, V396C and F397L - without rationale and a direct comparison with TMEM16F and a series of non-identifiable mutations. Where are the SERINC5 residues with respect to the structure? Why were they chosen as suitable mutations?
5. Figure 5 remains poorly rendered. There are no obvious routes for lipid flipping acquired from the analysis, with all lipids remaining in the same leaflet. Binding sites are shown in red densities, but there is no scale bar shown to know what period of the simulation the lipids remain bound to each site. A series of residues are annotated, but none of these relate to the residues tested in the previous figure. At what point in the simulation is the water channel apparent? All of the simulation? Just this snapshot? There are tools available in the community, e.g. HOLE, CHAP, CAVER, HOLLOW to test this more rigorously.
6. The Extended movies show domain rearrangement about the central TM helix, but there is no analysis performed to suggest that any access changes occur that would be illustrative of a transport mechanism.

This has the potential to be a nice story but the quality of the manuscript's presentation at present is limited and has not changed since I critiqued the work previously.

Reviewer #6 (Remarks to the Author):

Leonhardt and colleagues have submitted a substantial manuscript describing the structure of human SERINC3 and evidence that this protein restricts HIV infectivity by disrupting membrane asymmetry. I think this work is highly significant to the field and provides a substantial advance in understanding how SERINC3 influences HIV infectivity. This manuscript has been thoroughly and expertly reviewed and I have been asked to comment specifically on the evidence that hSERINC3/5 is a lipid transporter. The primary concern appears to be the observation that SERINC3 mutations that disrupt HIV restriction do not perturb scramblase activity in the reconstituted liposome system.

While the liposome-based scramblase assay can provide clear evidence that a purified protein is sufficient to scramble (or flip) phospholipid, these assays can also be prone to artifacts when the protein of interest is reconstituted in membranes that are not a good mimic of the native membrane. An instructive example is the observation that some GPCRs can constitutively scramble phospholipids when reconstituted in phosphatidylcholine/phosphatidylglycerol vesicles of the type used in this report (PMID 25296113). I am unaware of any evidence that these GPCRs can scramble lipid in their native membrane environment or that there is any biological significance to this observed scramblase activity. In addition, it was recently reported that the inclusion of cholesterol in the liposomes ablates the GPCR scramblase activity (PMID 35660161). This result suggests one potential consequence of denuding a membrane protein of its normal annular lipids, like cholesterol, is the provision of an artificial conduit for phospholipid movement between leaflets.

While the potential for in vitro artifact is an important caveat to keep in mind, the authors of this manuscript have a decided advantage in their studies because they have good evidence that SERINC3/5 disrupts membrane asymmetry in the native membrane environment and a strong correlation of this in vivo "scramblase" activity to the biological effect of inhibiting HIV infectivity. I was very impressed with the observation that the expression of constitutively active TMEM16F scramblase caused a near-equivalent inhibition of HIV infectivity as SERINC5. The logical conclusion is that SERINC3/5 are either scramblases or they regulate the activity of a scramblase.

It is also quite possible that the normal regulation of scramblase activity is lost in the reconstituted system and I think this is likely to be the case for the SERINC proteins. Most of the scramblases that have been characterized thus far, like TMEM16F or Xkr8, are constitutively "off" and require some activating event to turn them on. It is quite possible that the SERINC5 F397L and S328I mutations disrupt the normal regulatory transition from off to on without perturbing the lipid transport pathway. The artificial membrane in the reconstituted system may bypass normal regulation such that all of the SERINC3s as well as the SERINC5 mutants are constitutively turned on. However, in the absence of evidence that such a regulatory mechanism exists, the lack of correlation between in vitro scramblase activity and in vivo antiviral activity remains a concern and it seems appropriate to soften the emphasis of the conclusions.

My specific recommendations.

1. Remove the statement in the title that claims SERINC3 functions as a lipid transporter and replace it with something like "Structure of human SERINC3 and its role in HIV restriction through disruption of viral membrane asymmetry".
2. The last sentence of the abstract should also be changed to something like "We conclude that SERINC3s regulate membrane asymmetry and this activity directly correlates with loss of infectivity."
3. The final paragraph of the discussion should be softened to say our results strongly suggest that the SERINC3s are lipid scramblases and lipid flipping is strongly correlated..... It would make sense to describe caveats here – potential reasons why SERINC2 and the SERINC5 mutants exhibit scramblase activity in vitro but do not restrict HIV.

Replies to the Referees

NCOMMS-22-49012-T

CryoEM Structures of the Human HIV-1 Restriction Factor SERINC3 and its Role in Disrupting Virus Membrane Asymmetry

Reviewer #3 (Remarks to the Author):

The authors have improved their manuscript further by providing additional data, discussing the limit of the current studies, and revising the manuscript. I agree with the authors that there could be a number of reasons for the discrepancy between in vitro and in vitro flipping activities and authors provides a discussion for potential reasons for the discrepancy. I recommend publication of this manuscript to Nature communications.

Response: We certainly appreciate that the reviewer feels that the manuscript deserves publication in *Nature Communications*.

Reviewer #5 (Remarks to the Author):

I have previously reviewed this manuscript, when it was under consideration by [REDACTED]. I thank the authors for responding to my criticisms of the manuscript at this point. However, in terms of revisions to the manuscript, most notably the figures, very little has been done to address my concerns. I detail these in full below:

Response: We appreciate the careful reading and detailed suggestions by Reviewer 5 to improve the manuscript.

1. Figure 1 remains very raw in its presentation at present, I recommend:
 - a. removing the black backgrounds from the molecular graphics images.

Response: As suggested, Figure 1 was deleted and replaced with a complete revision according to the suggestions of Reviewer #5.

- b. Improving the resolution of the topology plot (Fig 1e).

Response: We agree and have increased the resolution.

- c. Aiming towards a presentation like shown in the figure below from the same journal: <https://www.ncbi.nlm.nih.gov/pmc/articles/PMC8654953/figure/Fig1/> i.e. (A) introduce to the reader what the proposed function of SERINC3 is, (B) provide a high quality illustration of the EM density. (C) Show the cartoon representation of SERINC beside the EM and detail key features. (D) Map out the topology. (E) Compare structural states (or save this for a later figure).

Response: There is no previous report in the literature suggesting that SERINC3s are lipid transporters that affect membrane asymmetry. Since this is a novel proposal, we are hesitant to present a model up front since it introduces bias. Instead, we prefer to engage in the process of scientific exploration in an unbiased way, by (1) presenting our cryoEM structures that suggest a similarity with the molecular design of non-ATP dependent lipid transporters, (2) functional studies in proteoliposomes and HIV-1 and MLV particles supporting our hypothesis and (3) conclude with the new final Figure 4 that presents our

hypothesis for lipid flipping via an alternating access mechanism. In this way, we are sharing our results and hypotheses with the scientific community in a way that we hope will generate follow-on studies in the field.

2. Figure 2 should be SI material. The best iterations of this type of figure detail each TM helix or feature one-by-one and also provide full details on the EM workflow, e.g.: SI Fig 2 from <https://www.ncbi.nlm.nih.gov/pmc/articles/PMC8654953/>
This is equivalent to Extended Figure 3 in this manuscript.

Response: As suggested, main text Figure 2 was moved to the supplementary data as Extended Data Figure 4.

3. Figure 3c-e require legends for each trace. There are resolution issues here, with the DPI for 3e being lower than the other traces.

Response: We agree and the resolution of the panels has been increased as requested.

4. Figure 4 then jumps to three SERINC5 mutants – S328I, V396C and F397L - without rationale and a direct comparison with TMEM16F and a series of non-identifiable mutations. Where are the SERINC5 residues with respect to the structure? Why were they chosen as suitable mutations?

Response: We agree and on page 9, we provide a more detailed background for the mutations and cite Pye *et al.*, who performed an extensive analysis of the functional effects of numerous mutations in SERINC5 and show their putative locations in the structure (Extended Data Figure 6i, j).

5. Figure 5 remains poorly rendered. There are no obvious routes for lipid flipping acquired from the analysis, with all lipids remaining in the same leaflet. Binding sites are shown in red densities, but there is no scale bar shown to know what period of the simulation the lipids remain bound to each site. A series of residues are annotated, but none of these relate to the residues tested in the previous figure. At what point in the simulation is the water channel apparent? All of the simulation? Just this snapshot? There are tools available in the community, e.g. HOLE, CHAP, CAVER, HOLLOW to test this more rigorously.

Response: Given our goal to accelerate publication and the extensive requests regarding our MD simulations, we decided to delete the MD simulations from our manuscript. The new Figure 4 focuses on the evidence supporting the hypothesis for an alternating access mechanism; i.e., the conformational states of full-length and the ICL4 deletion mutant of SERINC3 and the AlphaFold analysis that predicts conformational states similar to our cryoEM maps, with motions consistent with an alternating access mechanism. In this way, we are not speculating what the specific conformational states of the lipids are in achieving flipping between leaflets, which can be pursued by our lab and others in follow-on studies.

6. The Extended movies show domain rearrangement about the central TM helix, but there is no analysis performed to suggest that any access changes occur that would be illustrative of a transport mechanism.

Response: The motions indicated in the movies are more impactful than the superpositions shown in Figure 4 and are consistent with the conformational states of MurJ, which support an alternating access mechanism. Nevertheless, the specific protein lipid interactions are beyond the scope of our current work, and in fact are still not known for MurJ.

This has the potential to be a nice story but the quality of the manuscript's presentation at present is limited and has not changed since I critiqued the work previously.

Response: We understand and appreciate the reviewer's patience and their thorough review to improve our manuscript.

Reviewer #6 (Remarks to the Author):

Leonhardt and colleagues have submitted a substantial manuscript describing the structure of human SERINC3 and evidence that this protein restricts HIV infectivity by disrupting membrane asymmetry. I think this work is highly significant to the field and provides a substantial advance in understanding how SERINC3 influence HIV infectivity. This manuscript has been thoroughly and expertly reviewed and I have been asked to comment specifically on the evidence that hSERINC3/5 is a lipid transporter. The primary concern appears to be the observation that SERINC3 mutations that disrupt HIV restriction do not perturb scramblase activity in the reconstituted liposome system.

While the liposome-based scramblase assay can provide clear evidence that a purified protein is sufficient to scramble (or flip) phospholipid, these assays can also be prone to artifacts when the protein of interest is reconstituted in membranes that are not a good mimic of the native membrane. An instructive example is the observation that some GPCRs can constitutively scramble phospholipids when reconstituted in phosphatidylcholine/phosphatidylglycerol vesicles of the type used in this report (PMID 25296113). I am unaware of any evidence that these GPCRs can scramble lipid in their native membrane environment or that there is any biological significance to this observed scramblase activity. In addition, it was recently reported that the inclusion of cholesterol in the liposomes ablates the GPCR scramblase activity (PMID 35660161). This result suggests one potential consequence of denuding a membrane protein of its normal annular lipids, like cholesterol, is the provision of an artificial conduit for phospholipid movement between leaflets.

While the potential for in vitro artifact is an important caveat to keep in mind, the authors of this manuscript have a decided advantage in their studies because they have good evidence that SERINC3/5 disrupts membrane asymmetry in the native membrane environment and a strong correlation of this in vivo "scramblase" activity to the biological effect of inhibiting HIV infectivity. I was very impressed with the observation that the expression of constitutively active TMEM16F scramblase caused a near-equivalent inhibition of HIV infectivity as SERINC5. The logical conclusion is that SERINC3/5 are either scramblases or they regulate the activity of a scramblase.

It is also quite possible that the normal regulation of scramblase activity is lost in the reconstituted system and I think this is likely to be the case for the SERINC proteins. Most of the scramblases that have been characterized thus far, like TMEM16F or Xkr8, are constitutively "off" and require some activating event to turn them on. It is quite possible that the SERINC5 F397L and S328I mutations disrupt the normal regulatory transition from off to on without perturbing the lipid transport pathway. The artificial membrane in the reconstituted system may bypass normal regulation such that all of the SERINC3s as well as the SERINC5 mutants are constitutively turned on. However, in the absence of evidence that such a regulatory mechanism exists, the lack of correlation between in vitro scramblase activity and in vivo antiviral activity remains a concern and it seems appropriate to soften the emphasis of the conclusions.

My specific recommendations.

1. Remove the statement in the title that claims SERINC3 functions as a lipid transporter and replace it

with something like “Structure of human SERINC3 and its role in HIV restriction through disruption of viral membrane asymmetry”.

Response: We agree with the reviewer’s evaluation and have changed the title accordingly. We do have to adhere to the length restrictions and would suggest a shorter version of the suggested title:

“CryoEM Structures of the Human HIV-1 Restriction Factor SERINC3 and its Role in Disrupting Virus Membrane Asymmetry”

2. The last sentence of the abstract should also be changed to something like “We conclude that SERINC3s regulate membrane asymmetry and this activity directly correlates with loss of infectivity.”

Response: We agree and have changed the last sentence accordingly:

“We conclude that SERINC3s are lipid transporters, and we demonstrate that loss of membrane asymmetry is directly correlated with loss of infectivity.”

3. The final paragraph of the discussion should be softened to say our results strongly suggest that the SERINC3s are lipid scramblases and lipid flipping is strongly correlated..... It would make sense to describe caveats here – potential reasons why SERINC2 and the SERINC5 mutants exhibit scramblase activity in vitro but do not restrict HIV.

Response: We agree, but we also want to limit our speculations. As noted, our results do not indicate what the specific conformational states of the lipids are in achieving flipping between leaflets, which can be pursued by our lab and others in follow-on studies. For instance, a more rigorous MD analysis may suggest specific conformations and pathways for lipid transport, which can be tested by mutagenesis, crosslinking and spectroscopic studies (e.g., DEER EPR).